

# Implementation and Performance of Adaptive Mesh Refinement in the Ice Sheet System Model (ISSM v4.14)

Thiago Dias dos Santos[1], Mathieu Morlighem[2], Hélène Seroussi[3], Philippe Remy Bernard Devloo[1], and Jefferson Cardia Simões[4]

[1]Department of Structures, School of Civil Engineering, Architecture and Urban Design, University of Campinas - UNICAMP, Brazil
[2]Department of Earth System Science - University of California Irvine, CA, USA
[3]Jet Propulsion Laboratory, California Institute of Technology, Pasadena, CA, USA
[4]Polar and Climate Center, Geosciences Institute, Federal University of Rio Grande do Sul - UFRGS, Brazil

**Correspondence:** Thiago Dias dos Santos (santos.td@gmail.com)

**Abstract.** Accurate projections of the evolution of ice sheets in a changing climate require a fine mesh/grid resolution to correctly capture fundamental physical processes, such as the evolution of the grounding line, the region where grounded ice starts to float. The evolution of the grounding line indeed plays a major role in ice sheet dynamics, as it is a fundamental control on marine ice sheet stability. Numerical modeling of grounding line requires significant computational resources since the accuracy of its position depends on grid or mesh resolution. A technique that improves accuracy with reduced computational cost is the adaptive mesh refinement approach, AMR. We present here the implementation of the AMR technique in the finite element Ice Sheet System Model (ISSM) to simulate grounding line dynamics under two different benchmarks, MISMIP3d and MISMIP+. We test different refinement criteria: (a) distance around grounding line, (b) a posteriori error estimator, the Zienkiewicz-Zhu (ZZ) error estimator, and (c) different combinations of (a) and (b). We find that for MISMIP3d setup, refining 5 km around the grounding line, both on grounded and floating ice, is sufficient to produce AMR results similar to the ones obtained with uniformly refined meshes. However, for the MISMIP+ setup, we note that there is a minimum distance of 30 km around the grounding line required to produce accurate results. We find this AMR mesh-dependency is linked to the complex bedrock topography of MISMIP+. In both benchmarks, the ZZ error estimator presents high values around the grounding line. Particularly for MISMIP+ setup, the estimator also presents high values in the grounded part of the ice sheet, following the complex shape of the bedrock geometry. This estimator helps guide the refinement procedure such that AMR performance is improved. Our results show that computational time with AMR depends on the required accuracy, but in all cases, it is significantly shorter than for uniformly refined meshes. We conclude that AMR without an associated error estimator should be avoided, especially for real glaciers that have a complex bed geometry.

## 1 Introduction

The uncertainty in projections of ice sheet contribution to sea level rise in the next century remains large, primarily due to the potential collapse of the West Antarctic Ice Sheet, WAIS (Church et al., 2013; Jevrejeva et al., 2014; Ritz et al., 2015; DeConto



and Pollard, 2016). The collapse of WAIS is based on the Marine Ice Sheet Instability (MISI) hypothesis (Weertman, 1974; Mercer, 1978; Thomas, 1979). This hypothesis refers to ice sheets grounded below sea level on retrograde bedrock slopes (as seen in Figure 1), as is the case for many glaciers in WAIS (Fretwell et al., 2013). MISI states that the grounding line (GL), the region where the ice sheet starts to float (see Figure 1), cannot remain stable on such bedrock slopes (Schoof, 2007b; Katz

and Worster, 2010; Gudmundsson et al., 2012). Accordingly, the GL retreat on retrograde bedrock slopes causes increased ice discharge, which in turn leads to further GL retreat, resulting in a non-linear positive feedback. This self-sustaining GL retreat persists until a prograde bedrock slope is reached. Therefore, a change in climate or ocean can potentially force a large-scale fast migration of the GL inland (Schoof, 2007a; Favier et al., 2014; Seroussi et al., 2014b). Recently, the region of WAIS has experienced an increase in the intrusion of ocean warm deep water (Jacobs et al., 2011; Dutrieux et al., 2014) that probably

triggered the retreat of the GL observed over the last decades (Rignot et al., 2014; Christie et al., 2016; Kimura et al., 2016; Seroussi et al., 2017).

Modeling this positive feedback requires the coupling of different physical processes (ice sheet and ice shelf evolutions, GL migration, basal friction, etc.), and the accuracy of the results is highly dependent on the GL parameterization and the spatial discretization of the domain. Vieli and Payne (2005) compared the results of different ice sheet numerical models in terms of

GL migration, and found that the numerical results have a strong dependency on horizontal grid size. Analyzing the stability and dynamics of the GL on reverse bed slopes, Schoof (2007b) pointed out that high grid resolution in the GL zone is a critical element to obtain reliable numerical results. Two ice sheet model intercomparison projects later confirmed the GL dynamics dependency on spatial resolutions (Pattyn et al., 2012, 2013).

Several marine ice sheet models have employed different numerical schemes to overcome this mesh resolution requirement

at the GL with reduced computational cost: by imposing flux condition at GL position (Pollard and DeConto, 2009, 2012; Pattyn, 2017), by treating GL and basal friction with sub-grid or sub-element schemes (Feldmann et al., 2014; Leguy et al., 2014; Seroussi et al., 2014a) or by applying high spatial resolution only in the GL region with adaptive grid refinement (Durand et al., 2009; Goldberg et al., 2009; Gladstone et al., 2010; Gudmundsson et al., 2012; Cornford et al., 2013).

The grid or mesh adaptation technique allows to apply resources only where they are required, which is very useful in

transient simulations that include some discontinuity in the time-dependent solution (Devloo et al., 1987; Berger and Colella, 1989), as is the case for GL dynamics (the basal friction is only applied to grounded ice). This technique can be performed with two different methods: $r$-adaptivity and $h$-adaptivity methods (Oden et al., 1986). The $r$-adaptivity, also known as moving mesh method, moves progressively a fixed number of vertices in a given direction or region (Anderson et al., 1984, p. 533), while the $h$-adaptivity method, named in this work as adaptive mesh refinement (AMR), splits edges and/or elements, inserting

new vertices into the mesh where high resolution is required (Devloo et al., 1987; Berger and Colella, 1989). The performance of each of these methods depends on the problem for which they are applied. Vieli and Payne (2005) showed that models applying moving grid to track the GL movement perform better than fixed grid models. Since the position of the GL is explicitly defined in moving grids, Vieli and Payne (2005) noticed for this method a weak grid resolution dependency in comparison to fixed grid method, for which the GL position falls between grid points. Goldberg et al. (2009) obtained accurate solutions

with fewer resources solving the time-dependent Shelfy-Stream equations with the two different mesh adaptation techniques



mentioned above, moving mesh and AMR. Using a one-dimensional Shelfy-Stream model based on finite difference scheme, Gladstone et al. (2010) demonstrated that AMR and sub-grid parameterization for GL position could produce robust predictions of GL migration. Pattyn et al. (2012) found that moving grid methods tend to be the most accurate and AMR can further improve accuracy compared to models based on fixed grid. Cornford et al. (2013) implemented a block-structured AMR in

BISICLES, a three dimensional ice sheet model based on the finite volume method. They demonstrated that simulations with AMR are computationally cheaper and more efficient, even as the grounding line moves over significant distances. Jouvet and Gräser (2013) combined the Shallow Ice Approximation and the Shallow Shelf Approximation in an AMR numerical scheme involving a truncated Newton multigrid and finite volume method. Through MISMIP3d experiments (Pattyn et al., 2013), they highlighted the relevance and efficiency of AMR in terms of computational cost when high resolution (∼100 m) is necessary

to reproduce GL reversibility. Recently, Gillet-Chaulet et al. (2017) implemented an anisotropic mesh adaptation in the finite element ice flow Elmer/Ice (Gagliardini et al., 2013). Based on MISMIP+ experiment (Asay-Davis et al., 2016), they showed that combining various variables (ice thickness, basal drag, velocity, etc.) in an estimator allowed to reduce the number of mesh vertices by more than one order of magnitude compared to uniformly refined meshes, for the same level of numerical accuracy.

Here, we implement the AMR technique for unstructured meshes in the parallel finite element Ice Sheet System Model

(ISSM v4.14, Larour et al., 2012). The AMR capability in ISSM relies on two different and independent mesh generators: Bamg, a bidimensional anisotropic mesh generator developed by Hecht (2006), and NeoPZ, a finite element library developed by Devloo (1997). ISSM's architecture is based on the Message Passing Interface (MPI), where models are run in a distributed memory scheme. Our AMR implementation minimizes MPI communications, avoiding overheads and latencies. Since refinement criteria are crucial to AMR performance (Devloo et al., 1987), we implement different criteria based on: (a) the distance

to the grounding line, (b) the ZZ error estimator (Zienkiewicz and Zhu, 1987), and (c) different combinations of (a) and (b). To analyze the performance of AMR, we run two benchmark experiments: MISMIP3d (Pattyn et al., 2013) and MISMIP+ (Asay-Davis et al., 2016). We compare AMR results from both Bamg and NeoPZ with uniformly refined meshes in terms of GL position and computational time.

This paper is organized as follows: in Section 2, we summarize the main features of ISSM's architecture, and the strategies

used to implement an efficient AMR technique for transient simulations. In Section 3, we describe both MIMISP3d and MISMIP+ experiments used to analyze the AMR performance, and in Section 4 we present the results in terms of GL position and processing time. We finish this paper with a discussion of the results and conclusions in Sections 5 and 6, respectively.

## 2 AMR implementation

### 2.1 ISSM architecture

Our AMR implementation is strongly based on the architecture of ISSM. We describe here the main ISSM features necessary to understand the AMR strategy. We refer to Larour et al. (2012) for a more detailed description of ISSM.

Several stress balance approximations are implemented in ISSM, including higher-order models (e.g., Blatter-Pattyn, Pattyn (2003), full-Stokes). The current AMR capability is supported for the 2-D vertically integrated Shallow-Shelf or Shelfy-Stream



Approximation (SSA, Morland, 1987; MacAyeal, 1989). The SSA is employed for both grounded and floating ice, so membrane stress terms (which are required to correctly model the GL dynamics, Schoof, 2007b) are included but all vertical shearing is neglected (Seroussi et al., 2014a). Here, the mesh used for the SSA equations is unstructured and relies on a Delaunay triangulation.

ISSM is designed to run in parallel in a distributed memory fashion by Message Passing Interface (MPI). When a model is launched, the entire mesh is spatially partitioned over processing units or cores (CPU's), and data structures related to finite element method are built in each partition. All physical entities that vary in space (ice viscosity, ice thickness, surface, velocities, etc.) are kept within the elements.

MPI communications between the partitions (CPU's) are performed to assemble the global stiffness matrix and load vector,
as well as during the solution update in the elements once the system of equations is solved. The advantage of MPI is its ability to handle larger models (i.e. for continental scale simulations) in several cores and nodes on a cluster. Its disadvantage is the cost in the communications between the partitions.

## 2.2  Bamg and NeoPZ

The AMR technique in ISSM is implemented for unstructured meshes and triangular elements. Here are some short descriptions
of the meshers Bamg and NeoPZ.

Bamg (Hecht, 2006) is a bidimensional mesh generator based on Delaunay-like method (Hecht, 2005). This mesh generator is embedded in ISSM for static anisotropic mesh adaptation (Morlighem et al., 2010). Here, we extend Bamg capabilities for dynamic adaptation (AMR). The refinement in Bamg is carried out by specifying the desired resolution on the vertices. To reach the desired resolution, Bamg's algorithm splits triangle edges and inserts new vertices in the mesh (Hecht, 2006). The
algorithm keeps new vertices and connectivities unchanged as much as possible compared to the previous mesh (Hecht, 2005). This procedure reduces the numerical perturbations when the solutions are interpolated into the new mesh (see Section 2.3). Regions of different resolutions are linked by a transition zone, where the element sizes are changed gradually. The spatial extent of this transition zone is also specified in the Bamg's algorithm. An example of an adapted mesh using Bamg is shown in Figure 5.

NeoPZ (Devloo, 1997) is a finite element library dedicated to high adaptive techniques (Calle et al., 2015). In NeoPZ's data structure, each element is refined into 4 topologically similar elements, whose resolutions are half of the refined element. To avoid hanging vertices (Calle et al., 2015), some elements are divided in specific ways such that any two elements in the mesh may have a vertex or an entire edge in common, or no vertices in common (Szabó and Babuška, 1991, p. 81). In this sense, all meshes refined by NeoPZ are nested, i.e., vertices and connectivities from the coarse mesh are kept fixed during all simulation
time. This characteristic does not introduce any perturbation during the interpolation process (see Section 2.3). The AMR with NeoPZ is given by specifying the level of refinement, i.e., how often elements are refined. Therefore, the transition zone, which links regions of different resolutions, is generated stepwise through resolutions dictated by levels of refinement. Figure 5 shows examples of adapted meshes using NeoPZ.



Here, we describe the algorithm to couple ISSM to Bamg and NeoPZ as well as the refinement criteria usage (Sections 2.3 and 2.4).

## 2.3 Parallel strategy

The solution sequence for transient simulation with AMR is summarized in Figure 2. Details of AMR processes are itemized
in Algorithm 1. In Figure 2, the AMR is the last step to be executed for a given time step. This is an explicit approach, where a new adapted mesh is built for a given solution. In Algorithm 1, all processes involved to perform the AMR in ISSM are executed in the step 'e', the remeshing core. Step 'e.1' executes the mesh adaptation (refinement or coarsening of elements) and the other steps ('e.2' to 'e.5') prepare the adapted mesh, data structures and solutions for the next simulation time step.

Bamg and NeoPZ perform the AMR (step 'e.1', Algorithm 1) in serial, considering the entire mesh. In our implementation,
only one partition (which CPU rank is #0) keeps the Bamg or NeoPZ entire mesh, and is responsible to execute the AMR process.

Our AMR implementation keeps the number of partitions constant during all simulation time. The number and distribution of elements into the partitions varies every time AMR is called, since the mesh partitioning process (step 'e.2', Algorithm 1) generates partitions with similar number of elements. This process avoids memory imbalance among the CPU's and overheads
during the solver phase (Larour et al., 2012).

Each time remeshing is performed, the solutions and all data fields are interpolated from the old mesh onto the new mesh. This step is executed in parallel, where each CPU interpolates the solutions just on its own partition (step 'e.4', Algorithm 1). The construction of new data structures and the adjustment of solutions (steps 'e.3' and 'e.5', respectively, Algorithm 1) are also executed in parallel, as is the computation of the refinement criteria (see Section 2.4).

All MPI communications in the remesh core (step 'e', Algorithm 1) are minimized to avoid overheads when large models are run. In the interpolation process, for example, all relevant fields are collected by CPU #0 in one single vector structure in such a way that only one MPI broadcast is called. This approach is based on the fact that, in general, it is more efficient to perform few large MPI messages instead of carrying out many smaller ones (Reinders and Jeffers, 2015, p. 327).

## 2.4 Refinement criteria

The grounding line dynamics is implemented in ISSM through an implicit level set function, $\phi_{gl}$, based on a hydrostatic floatation criterion (Seroussi et al., 2014a):

$$\phi_{gl} = H - H_f, \tag{1}$$

where $H$ is the ice thickness and $H_f = -b(\rho_w/\rho_i)$ is the floating height, being $\rho_i$ the ice density, $\rho_w$ the ocean density and $b$ the bedrock elevation (negative if below sea level). Figure 1 illustrates the GL position in a vertical plane view of a marine ice





sheet. The position of GL is defined as:

$$\begin{cases} \phi_{gl} < 0 : & \text{ice is floating} \\ \phi_{gl} > 0 : & \text{ice is grounded} \\ \phi_{gl} = 0 : & \text{grounding line position} \end{cases}, \tag{2}$$

The performance of AMR depends on the refinement criterion (Devloo et al., 1987). We implement the three following criteria:

(a) Element distance to GL, $R_{gl}$;

(b) ZZ error estimator for deviatoric stress tensor, $\tau$ and ice thickness, $H$;

(c) Different combinations of (a) and (b).

The criterion (a) is based on a heuristic approach commonly applied (Goldberg et al., 2009; Gudmundsson et al., 2012; Cornford et al., 2013). The second criterion, (b), is based on the fact that high changes in the gradients in the velocity field

(therefore, in the deviatoric stress tensor, $\tau$) and ice thickness, $H$, are expected to be present near the grounding line. Criterion (c) extends and merges the features of the other two previous criteria. We define the AMR criterion used based on binary flags $\theta$ ($= 0$ or $1$) such that:

$$\begin{cases} \theta_{gl} = 1 : & \text{use element distance to GL} \\ \theta_{\tau} = 1 : & \text{use ZZ error estimator for } \tau \\ \theta_H = 1 : & \text{use ZZ error estimator for } H \end{cases}. \tag{3}$$

We propose Algorithm 2, inspired by Devloo et al. (1987), to execute the refinement and coarsening processes under different

criteria (AMR core, step 'e.2' in Algorithm 1). The first 3 steps in Algorithm 2 compute the criterion according to the binary flags, $\theta$, defined above. These steps are performed in parallel. Step '4' verifies, for each element in the mesh, if it should be refined: its distance to GL and its ZZ errors are compared with prescribed limits (thresholds). The element is refined if at least one of the thresholds is exceeded, so long as its level of refinement is less than the maximum level chosen. This logical operation is performed by the operator "or" in the statement "if" in step '4'. Once an element is refined, it is identified as a

group. Step '5' verifies for each group if it should be coarsened. To be coarsened, a group should meet all thresholds; the logical operator used in this case is "and" (statement "if" in step '5'). Algorithm 2 has two sets of thresholds (shown with $max$), for elements and for groups of elements. For the algorithm to work properly, these sets of thresholds should be different, following Devloo et al. (1987).

### 2.5 ZZ error estimator

The generic form of the ZZ (Zienkiewicz and Zhu, 1987) error estimator $\epsilon(e)$ for a given element $e$ is:

$$\epsilon(e) = \left[ \int_{\Omega_e} (\nabla u^* - \nabla u)^2 \, d\Omega_e \right]^{1/2}, \tag{4}$$





where $\Omega_e$ is the domain of the element $e$, $\nabla u$ is the gradient of the finite element solution $u$ and $\nabla u^*$ is the smoothed reconstructed gradient, calculated on the element $e$ as:

$$\nabla u^* = \sum_{i=1}^{s} \psi_i \nabla u_i^*, \tag{5}$$

and

$$\nabla u_i^* = \frac{1}{W_i} \sum_{j=1}^{k} w_j \nabla u_j, \tag{6}$$

where $\psi_i$ is the $i$th P1 Lagrange shape function on element $e$, $s$ is the number of shape functions of the element $e$ (here, $s = 3$), $j$ is the $j$th element connected to the vertex $i$, $k$ is the number of elements connected to vertex $i$, $w_j$ is the weight relative to the element $j$ and $W_i$ is the sum of all weights for the vertex $i$. Here, the weights $w$ are defined as the geometric area of the triangular elements. We implement the ZZ error estimator for the ice thickness ($u = H$) and we extend the estimator for the deviatoric stress tensor ($\tau$), written in a vectorized form, i.e., for SSA we have $\nabla u \rightarrow \boldsymbol{\tau} = (\tau_{xx}, \tau_{yy}, \tau_{xy})^{\mathsf{T}}$. See Section 2.4 and Algorithm 2 for how these error estimates are combined.

## 3 Numerical experiments

We run 2 different benchmark experiments to evaluate the adaptive mesh refinement capabilities based on the MISMIP3D (Pattyn et al., 2013) and MISMIP+ (Asay-Davis et al., 2016) experiments. The following subsections describe briefly each setup. More details can be found in the respective references. All experiments are performed using the vertically integrated Shelfy-Stream Approximation equations (SSA, Morland, 1987; MacAyeal, 1989).

### 3.1 MISMIP3d setup

The MISMIP3d domain setup is rectangular, and extends from 0 to 800 km in the $x$ direction and from 0 to 50 km in the $y$ direction. The bed elevation ($b$) varies only in the $x$ direction, as follows:

$$b(x,y) = -100 - x. \tag{7}$$

Boundary conditions are applied as follows: a symmetric condition at $x = 0$ so that ice velocity is equal to zero, a symmetric condition at $y = 0$ (which represents the centerline of the ice stream), and free slip condition at $y = 50$ km so that the flux through these surfaces is zero. Water pressure is applied at the ice front at $x = 800$ km.

The ice viscosity, $\mu$, is considered to be isotropic and to follow the Glen's flow law (Cuffey and Paterson, 2010):

$$\mu = \frac{B}{2\dot{\epsilon}_e^{\frac{n-1}{n}}}, \tag{8}$$

where $B$ ($= A^{1/n}$, being $A$ the Glen's flow law factor) is the ice viscosity parameter, $\dot{\epsilon}_e$ is the effective strain rate and $n = 3$ the Glen's exponent. The ice viscosity parameter, $B$, is uniform and constant over the domain and the time, and its value is



equal to $2.15 \times 10^8 \, \mathrm{Pa\,s}^{-1/3}$. A non-linear friction (Weertman) law is applied on grounded ice:

$$\boldsymbol{\tau}_b = C \, |\boldsymbol{u}_b|^{m-1} \, \boldsymbol{u}_b, \tag{9}$$

where $\boldsymbol{\tau}_b$ is the basal shear stress, $\boldsymbol{u}_b$ is the basal sliding velocity, $C$ is the friction coefficient, and $m = 1/3$ is the sliding law exponent. The basal friction coefficient, $C$, is also spatially uniform for all grounded ice, and equal to $10^7 \, \mathrm{Pa\,m}^{-1/3}\mathrm{s}^{1/3}$.

The experiments are run starting from an initial configuration with a thin layer (1 m) of ice and run until a steady state condition is reached, which occurs after 30,000 years. We compare the GL positions from different meshes at t=30,000 yr.

We investigate the sensitivity of the AMR for which the refinement method is based on the element distance to the GL, $R_{gl}$ (criterion (a), Section 2.4). For comparison analysis, three different distances are used for the highest refinement level: $R_{gl} = 5$, 10 and 15 km. These different meshes are labeled as R5, R10 and R15, respectively. The distance $R_{gl}$ refers to the region with

the highest level of refinement. For example, R5 means that 5 km on both sides of the GL (upstream and downstream) are refined with the highest level. Table 1 summarizes the criteria used for all experiments. The coarse mesh, that has a resolution of 5 km, is used as an initial mesh for all simulations and mesh generators (Bamg and NeoPZ). To analyze the convergence, we refine the coarse mesh $1\times$, $2\times$ and $3\times$. These 3 levels of refinement are applied to both uniform and adaptive meshes, and correspond to elements with 2.5, 1.25 and 0.63 km resolution, respectively. Table 2 presents the correspondence between level

and resolution at the GL used in the experiments.

We also investigate the sensitivity of the AMR to GL parameterization into the elements (Seroussi et al., 2014a). Three sub-element parameterizations are tested: no sub-element parameterization (NSEP), sub-element parameterization 1 (SEP1) and sub-element parameterization 2 (SEP2). In the NSEP method, each element of the mesh is either grounded or floating and the grounding line position is defined as the last grounded point. In SEP1 and SEP2 methods, the grounding line position is

located anywhere within an element and defined by the implicit level set function, $\phi_{gl}$, which is based on the floating condition (see Section 2.4). The difference between SEP1 and SEP2 is how each one of these methods computes the basal friction to match the amount of grounded ice in the element. In the SEP1 approach, the basal friction coefficient ($C$) is reduced as $C_g = C \, A_g/A$, where $C_g$ is the new basal friction coefficient for the element partially grounded, $A_g$ is the area of grounded ice of this element and $A$ is the total area of the element. In the SPE2 technique, the basal friction is integrated (in the sense of the finite element method) only on the part where the element is grounded. This is done by changing the integration area from

the original element to the grounded part of the element. We refer to Seroussi et al. (2014a) for a complete description of these sub-element parameterizations.

### 3.2 MISMIP+ setup

The MISMIP+ domain is also rectangular, whose dimensions are: $0 \le x \le 640$ km and $0 \le y \le 80$ km. The bed elevation is

defined as follows:

$$b(x,y) = \max\left(b_x(x) + b_y(y), b_{deep}\right). \tag{10}$$





with:

$$b_x(x) = b_0 + b_2 \left(\frac{x}{\bar{x}}\right)^2 + b_4 \left(\frac{x}{\bar{x}}\right)^4 + b_6 \left(\frac{x}{\bar{x}}\right)^6, \tag{11}$$

and

$$b_y(y) = \frac{d}{1 + \exp\left[-2(y - L_y/2 - w_c)/f_c\right]} + \frac{d}{1 + \exp\left[2(y - L_y/2 + w_c)/f_c\right]}, \tag{12}$$

where $b_{deep} = -720$ m, $b_0 = -150.0$ m, $b_2 = -728.8$ m, $b_4 = 343.91$ m, $b_6 = -50.57$ m, $\bar{x} = 300$ km, $d = 500$ m, $L_y = 80$ km, $w_c = 24$ km, $f_c = 4$ km. Figure 3 shows the bed elevation calculated with Equations 10, 11 and 12.

The ice is isothermal and the ice viscosity parameter, $B$, is equal to $1.1642 \times 10^8 \, \mathrm{Pa\,s}^{-1/3}$ (uniform and constant over the domain and the time). The boundary conditions are similar to MISMIP3d. The friction model used here is a power law, Eq. (9), with a coefficient, $C$, equal to $3.160 \times 10^6 \, \mathrm{Pa\,m}^{-1/3}\mathrm{s}^{1/3}$ (spatially uniform for all grounded ice) and sliding law exponent, $m$, equal to $1/3$.

We run the experiments starting from an initial configuration with a 1 m thick layer of ice, and run the simulations until a steady state condition is reach (after 20,000 years). The GL positions are compared at t=20,000 yr.

To investigate further the sensitivity of the GL position to the refinement distance, $R_{gl}$, we choose distances with the highest refinement level equal to $R_{gl} = 5$, 15 and 30 km, with meshes labeled as R5, R15 and R30, respectively (see Table 1). As for MISMIP3d simulations, these distances refer to the region around the GL with the highest level of refinement. The same coarse mesh with a resolution of 4 km is used for Bamg and NeoPZ, and it is refined up to four times for both adaptive and uniform refinement approaches. The respective resolutions for the four refinement levels are 2, 1, 0.5 and 0.25 km. Table 2 summarizes the levels and the respective resolutions at the GL. All MISMIP+ simulations are performed with sub-element parameterization type I, SEP1 (Seroussi et al., 2014a).

It is important to emphasize that the MISMIP+ bed elevation (Figure 3) is calculated directly in the code every time AMR is called (step 'e.5', Algorithm 1). This procedure avoids excessive smoothing of the complex bedrock topography in the refined region.

## 4 Results

For a given problem, the results from an AMR mesh should be as close as possible (within an acceptable tolerance) to those obtained with an uniformly refined mesh, for the same level of refinement (element resolution) in both meshes. This comparison is an indicator of the AMR performance to that given problem. Since Bamg and NeoPZ adapt the mesh in different ways, it is important to analyze how their differences impact the numerical solutions. Therefore, we first compare the results from the adaptive and uniform meshes using both Bamg and NeoPZ for the MISMIP3d and MISMIP+ experiments, and then we assess the time performance of the AMR in comparison with uniformly refined mesh.





### 4.1 GL position comparison

#### 4.1.1 MISMIP3d setup

Figure 4 presents the GL positions for different AMR meshes and sub-element parameterizations as a function of element resolutions. The refinement criterion used is the element distance to the GL, $R_{gl}$ (see Table 1 and Section 2.4). GL positions

obtained with uniformly refined meshes are also shown in Figure 4. For NSEP, GL position varies between 200 km and 520 km for meshes L0 (coarse mesh) and L3 (level of refinement equal to 3), respectively. For these same meshes, GL position varies between 620 km and 600 km for SEP1, and 550 km and 600 km for SEP2.

We note that AMR meshes with NeoPZ produces GL positions that are very similar to the ones produced with uniformly refined mesh. This holds for all sub-element parameterizations. AMR with Bamg is more sensitive to NSEP, for which GL

positions depend on the element distance to GL ($R_{gl}$) used, especially for the lower refinement level (level equal to 1). Despite this, GL positions from AMR with Bamg are in agreement with uniformly refined meshes for SEP1 and SEP2.

#### 4.1.2 MISMIP+ setup

The MISMIP+ bed topography (see Section 3.2 and Figure 3) is designed to introduce a strong buttressing on the ice stream from the confined ice shelf. It is therefore expected that the results are more sensitive to the mesh refinement compared to

simpler bedrock descriptions (e.g. MISMIP3d), since refining the mesh also improves the resolution of the bedrock geometry (see Section 3.2).

Figure 5 presents the coarse mesh and 3 examples of adaptive meshes obtained with Bamg and NeoPZ and different criteria: element distance to GL, $R_{gl}$ ($= 5$ km, R5) and error estimator ZZ (see Table 1). The figure also shows the GL positions obtained with these meshes. We can note that, using the same criterion based on the element distance to GL (meshes R5), Bamg and

NeoPZ produce different meshes, as expected. For Bamg, the transition zone between the lowest and highest resolution is smoother than NeoPZ's mesh, since the resolutions in NeoPZ are obtained stepwise by nested elements. At the center of the domain (y=40 km), the GL position varies by 10 km between the coarse mesh and the adaptive meshes (for both Bamg and NeoPZ). When the ZZ criterion is used, the GL positions differ by 17 km in comparison with the one obtained from the coarse mesh.

Figure 6 presents a set of results for the grounding line position as a function of mesh resolution. AMR mesh-dependency is clear in Figure 6. For AMR with NeoPZ, GL positions obtained with AMR R5 are below the ones produced by AMR R15 and AMR R30. Marginally AMR R15 and AMR R30 produce the same GL positions. For AMR with Bamg, AMR R5 and AMR R15 do not improve the position of the GL as the resolution increases. We can note the differences of GL positions from AMR (with both Bamg and NeoPZ) and from uniformly refined meshes are higher in comparison to MISMIP3d experiment.

To investigate the possible causes of this AMR mesh-dependency, we perform the AMR using the ZZ error estimator calculated for the deviatoric stress tensor, $\tau$ (Table 1). The GL positions obtained with AMR ZZ is presented in Figure 6, for the mesh generator NeoPZ only. We observe that GL positions with AMR ZZ are closer to the ones obtained with uniform meshes, for all mesh resolutions. To understand this AMR ZZ result, we plot the spatial distributions of the ZZ error estimator for the





coarse and adaptive meshes (using NeoPZ), as illustrated in Figure 7 (see also the movies in the supporting information). The ZZ error values are normalized between 0 and 1. For the coarse mesh, we see in Figure 7 that the error estimators calculated for the deviatoric stress tensor and the ice thickness present high values around the GL. In particular, for the ice thickness, the estimator also presents high values in the grounded part of the marine ice sheet, following the high gradient in the side

walls of the bed topography (see Figure 3). For AMR R5 meshes, there are high ZZ error values around the refined region. This is not observed when the refinement criterion used is the ZZ estimator (AMR ZZ, see Table 1), as expected. Using the ZZ criterion induces an equalization in the spatial distribution of the estimated errors, improving the solutions (e.g. GL position, see Figure 6). In terms of performance, AMR ZZ generates fewer elements than AMR R30. At the end of the experiment and for a level of refinement equal to 4 (resolution equal to 250 m), AMR R30 mesh has 464,712 elements, while AMR ZZ mesh

has 24,428 elements (i.e., only ∼5% of the number of elements in AMR R30).

## 4.2 AMR time performance

To analyze the AMR performance in terms of computational time, we run the experiment Ice1r of MISMIP+ (Asay-Davis et al., 2016). The experiment starts from the steady state condition and runs forward in time for 100 years with a basal melt rate applied. The time step is equal to 1 yr. Although the time step should be around ∼0.2 yr for the highest resolution mesh

(to fulfil the CFL condition), the simulations here do not present any numerical instabilities. The non-linear SSA equations are solved using the Picard scheme and the Multifrontal Massively Parallel sparse direct Solver, MUMPS (Amestoy et al., 2001, 2006).

We initialize the models using the steady state solution obtained with the coarse mesh. This procedure emulates a common modeling practice: a coarse mesh is initialized with observed fields and the AMR is carried out in the model during the

forcing (experiment) scenarios. It is important to note that, in the sense of the MISMIP+, both the steady state and the forcing experiments should be carried out using the same AMR mesh. The purpose of the experiments here is only to assess the processing time with AMR. All the experiments are run in parallel (16 cores) in a 2x Intel Xeon E5-2630 v3 2.40 GHz with 64 GB of RAM.

Table 3 presents GL positions obtained with different meshes at the end of experiment Ice1r, and the computational time

and number of elements required for each mesh, as well as the criterion used. The levels of refinement are labeled as 'L#', e.g., L3 means level 3 (see Table 2). Considering the GL position obtained from the highest resolution mesh (L4 uniform) as a reference result, we compare the computational cost using uniform and AMR meshes to obtain the same result within a deviation of 1 km (i.e. 394 ± 1 km). In Table 3, only L3 meshes produce this required accuracy. L3 uniform mesh has at least 3.5× more elements than the AMR L30 meshes, which represents a computational time 2.5× higher in comparison with the

adaptive approach. In terms of refinement criteria, AMR ZZ generates half the number of elements in comparison to AMR R30, which means virtually at least half of computational time. Comparing AMR ZZ and L3 uniform, the computational time using the adaptive mesh is at least 4× less. The performance of Bamg and NeoPZ is similar, once the ratio of computational time and number of elements is virtually equal for both packages.



## 5   Discussion

In this work, we describe the implementation of an adaptive mesh refinement approach in the Ice Sheet System Model (v4.14) as well as the performance of our implementation in terms of grounding line position and simulation time. We investigate the adaptive meshes performance using a criterion commonly applied by the community (distance to GL, Durand et al., 2009; Gold-

berg et al., 2009; Gudmundsson et al., 2012; Cornford et al., 2013), and we compare with an error estimator (ZZ, Zienkiewicz and Zhu, 1987) based on the a posteriori analysis of the transient solutions (deviatoric stress tensor and ice thickness).

We rely on two different mesh generators, Bamg (Hecht, 2006) and NeoPZ (Devloo, 1997) that have different properties. It is therefore expected that their solutions are not identical. This explains the difference observed in GL positions compared to uniform meshes for the 3 sub-element parameterizations (e.g., MISMIP3d setup, Figure 4).

NeoPZ generates nested meshes, which reduces errors in the interpolation step, what is useful to assess AMR performance in comparison to uniformly refined mesh. Bamg's algorithm works differently: the fact that some vertices positions change produces at least two side effects: (1) induced errors in the interpolation process; (2) positive or negative impact on the convergence of the solutions. The weight of the first side effect can be reduced using higher element distance to GL ($R_{gl}$), for which the highest resolution is applied, and increasing the length of the transition zone between fine and coarse elements.

Higher-order interpolative scheme can be also used, as pointed out by Goldberg et al. (2009), to avoid solution diffusion. In ISSM, the interpolation scheme is carried out by P0 and P1 Lagrange polynomials, and we suspect these are enough for our AMR procedure. The weight of the second side effect depends on the problem considered. We suspect that for GL dynamics this effect has overall a positive impact, once updating vertex positions is somewhat similar to the moving mesh technique, although the GL is not explicitly defined in our approach as in other studies (e.g., Vieli and Payne, 2005; Goldberg et al., 2009).

This argument is based on the results shown here, for both MISMIP3d and MISMIP+ setup. Some mesh packages and finite element libraries related to NeoPZ are: Deal II (Bangerth et al., 2007), Hermes (Šolín et al., 2008), libMesh (Kirk et al., 2006) and HP90 (Demkowicz et al., 1998). Mesh generators related to Bamg are: MMG (Dapogny et al., 2014), Yams (Frey, 2001) and Gmsh (Geuzaine and Remacle, 2009). So, we expect that the results shown here in this work would be reproduced with these related packages.

The results from MISMIP3d suggest that independently of the sub-element parameterization and refinement level, refining elements within a 5 km region with highest resolution around GL is enough to generate solutions similar to the ones produced by uniform meshes. This holds for Bamg and NeoPZ (Figure 4). Cornford et al. (2013) presented similar results for SSA equations through BISICLES, an AMR finite volume-based ice sheet simulator. They concluded that refining 4 cells on either sides of the GL was enough to achieve results similar to uniform meshes for all levels of refinement.

For MISMIP+, a minimal distance of 30 km for the highest resolution around GL is necessary to accurately capture the GL position (Figure 6). Nevertheless, there is a residual between GL positions from AMR and uniform meshes. This AMR mesh-dependency can be explained by the bed topography of MISMIP+ (Figure 3): the high gradient in the side walls induces numerical errors on the gradients of the velocity field (deviatoric stress tensor, near GL) and ice thickness (on grounded ice), as illustrated by the spatial distribution of the a posteriori error estimator used here (Figure 7). For MISMIP3d setup, the highest





values of the error estimate concentrate only around the GL (not shown here), which explains why a few kilometers of high resolution near the GL improves the GL positions.

Since numerical errors are not only concentrated near the GL for the MISMIP+ setup, an error estimator may be more appropriate to guide the refinement and to reduce the error estimates along the domain, improving AMR performance. This
explains why a simple test with the AMR ZZ produces better convergence with much less elements than AMR based on the heuristic criterion (element distance to GL, Figure 6). Related works have used proxies of error estimators: Goldberg et al. (2009) used the jumps in strain rate between adjacent cells; Gudmundsson et al. (2012) used the second derivative of ice thickness; Cornford et al. (2013) used the Laplacian of the velocity field and Gillet-Chaulet et al. (2017) used the estimator proposed by Frey and Alauzet (2005), whose metric is based on a priori interpolation error calculated by the field's Hessian
matrix (second derivative). The ZZ used here is a true a posteriori error estimator based on the recovered gradient (Ainsworth and Oden, 2000, p. 3), widely used in the finite element community (Ainsworth et al., 1989; Grätsch and Bathe, 2005) and suitable to be implemented in ice sheet models, or the ones based on finite volumes or finite differences. As the MISMIP+ bed geometry is more realistic than MISMIP3d, we can expect a similar result for real glaciers, i.e., high numerical errors present in regions not necessarily adjacent to the GL.

Further analysis with ZZ or another error estimator should be developed to improve the AMR criterion used in ice sheet modeling. An important issue to be investigated is the interpolation of real bed topography directly from a dataset every time AMR is carried out. This interpolation increases bed resolution according to mesh adaptation, which reduces the smoothness of the bed in the model (since real beds are not necessarily smooth). The reduction of the bed smoothness induces some numerical errors and counterbalances the effect of mesh adaptation, increasing AMR mesh-dependency. Real bed topographies should
be analyzed in benchmark models as well as in real ice sheet domains. Our current AMR implementation interpolates the bed elevation from the coarse (initial) mesh, except for the MISMIP+ experiment, for which we hard-coded the calculation of the bed topography directly in the code (in this case, AMR reduces the smoothness of the bed in the model, but as there is no noise, the numerical error based on the ZZ error estimator for the ice thickness is reduced). The interpolation from a dataset will be implemented in ISSM in the future. Based on this discussion and the results shown in this study, we recommend AMR with the
combination of the heuristic criterion (using a minimal distance, e.g., 5 km) with an associated error estimator. In other words, only imposing fine mesh resolution near the GL does not ensure that the GL position is correctly captured. Tests varying AMR parameters (distance to GL, maximum thresholds for the error estimator, level of refinement, etc.) should be carried before any ice sheet simulation to optimize AMR performance in terms of both solutions and computational time.

The grounding line zone is not the only place where AMR can be applied. Ice front and calving dynamics (Todd et al.,
2018) as well as shear margins in ice streams (Haseloff et al., 2015) are examples for which adaptive meshes can improve numerical solutions with reduced computational efforts. In ISSM, the AMR can be applied to these regions through extension of Algorithm 2. Other experiments (not shown here) testing the AMR to refine the ice front region show promising results (Santos et al., 2018).

Our AMR performance analysis shows that the computational time in AMR simulations reaches up to one order of magnitude
less in comparison to models based on uniform meshes. Computational time and solution accuracy of AMR depend on the





refinement criterion used, and even with a hundred of elements generated (e.g., meshes AMR R30), the computational time is satisfactory. This is observed for both NeoPZ and Bamg. Further analysis should be carried out to check the performance in higher computational scale (thousand of elements), but the results presented in this study suggest that our AMR implementation strategy is adapted to the modeling questions being investigated. Our AMR computation time compares to Cornford et al.

(2013), in which AMR simulations spend ∼1/3 of CPU time needed in simulations performed by uniform meshes.

## 6 Conclusions

We implemented here dynamic AMR into ISSM and tested its performance on two different experiments with different refinement criteria. The comparison between Bamg and NeoPZ shows that they present similar performance, and the choice of which to be used is up to the user. Moreover, users using Bamg (or similar mesh generator) should pay attention in the minimal

extension of the transition zone to reduce numerical errors (e.g. in the interpolation step). NeoPZ is more suitable with error estimators, as well as in AMR performance comparison. Based on the AMR mesh-sensitivity observed here, we conclude that AMR without an error estimator should be avoided, mainly in setups where bedrock induces strong buttressing. In real bedrock topographies, where small scale features may play an important role, an error estimator is suitable to guide the AMR. Further research should be carried out in order to evaluate AMR performance in real bed geometries. Our recommendation to improve

the AMR performance while minimizing computational effort is the combination of the heuristic criteria, applying a minimal distance around GL (e.g. 5 km), with an error estimator. The simple tests with the ZZ error estimator show a significant potential, mostly due to its simple implementation and performance. The AMR technique in ISSM can be extended to others physical processes such that the evolution of ice sheets and, consequently the sea level rise, can be accurately modeled and projected.

*Code availability.* The adaptive mesh refinement are currently implemented in the ISSM code for triangular elements. The code can be download, compiled and executed following the instructions avaiable on the ISSM website: https://issm.jpl.nasa.gov/download.

*Competing interests.* Authors declare no compenting interests.

*Acknowledgements.* This work was performed at the University of Campinas (UNICAMP) and Federal University of Rio Grande do Sul (UFRGS) with financial support from CNPq - Conselho Nacional de Desenvolvimento Científico e Tecnológico, Brasil, PhD scholarship $N^o$

140186/2015-8 - and at the University of California Irvine (UCI) under a contract with the National Aeronautics and Space Administration (NASA), Cryospheric Sciences Program ($N^o$ NNX14AN03G).



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





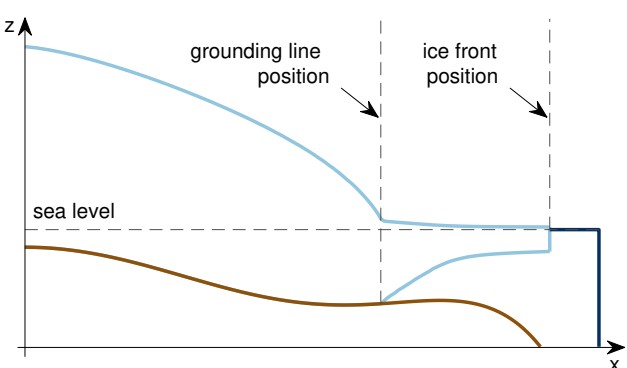

**Figure 1.** Vertical plane view of a marine ice sheet: marine ice sheet, ocean and **bed**. The position of the grounding line is implicitly defined by the level set function, $\phi_{gl}$, based on a hydrostatic floatation criterion (Seroussi et al., 2014a).



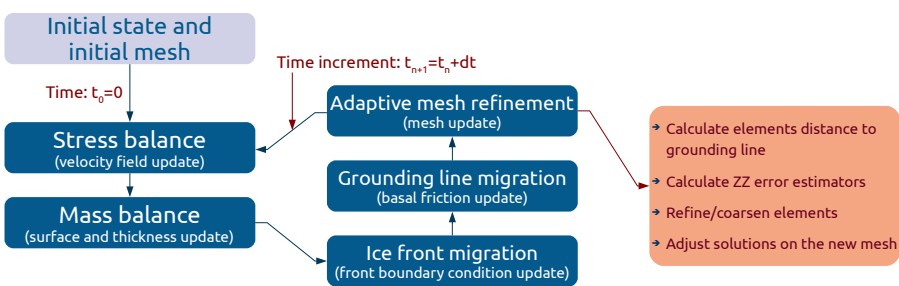

**Figure 2.** Solution sequence for ice sheet transient simulation with adaptive mesh refinement.





---

**Algorithm 1** Transient simulation with AMR

---

1. set initial solution state and initial coarse mesh

2. while $t_n \leq t_{max}$ do:

    a. call stress balance core (diagnostic)

    b. call thickness balance core (prognostic)

    c. call ice front migration core (level set adjustment)

    d. call grounding line migration core (hydrostatic adjustment)

    e. call remesh core (AMR)

        e.1. call AMR core (refine/coarsen mesh, Bamg or NeoPZ, serial in CPU #0)

        e.2. call mesh partitioning (over all CPU's, serial)

        e.3. build new data structures (all CPU's, parallel)

        e.4. interpolate solutions (all CPU's, parallel)

        e.5. call geometry adjustment core (all CPU's, parallel)

    f. time increment $t_{n+1} = t_n + dt$

3. post processing

---





---

**Algorithm 2** AMR core: refinement criteria calculation, refinement and coarsening processes. $e$ = element. $g$ = group of elements that are nested and derived from a refined element. $L(e)$ = level of refinement of the element $e$. $L^{max}$ = maximum level of refinement. $R^{max}$ = maximum threshold for element/group distance to grounding line. $\epsilon^{max}$ = maximum threshold for element/group error estimator (thickness/deviatoric stress). $\theta$ = binary flags that define the criterion to use.

---

1. if $\theta_{gl} = 1$, then **compute** the element and group distances to the grounding line, $R_{gl}(e)$ and $R_{gl}(g)$.

2. if $\theta_\tau = 1$, then **compute** the element and group deviatoric stress error estimators, $\epsilon_\tau(e)$ and $\epsilon_\tau(g)$.

3. if $\theta_H = 1$, then **compute** the element and group thickness error estimators, $\epsilon_H(e)$ and $\epsilon_H(g)$.

4. for each element $e$ such that $L(e) < L^{max}$, do:

    if $\left[R_{gl}(e) < \theta_{gl} \cdot R_{gl,e}^{max}\right]$ or if $\left[\theta_\tau \cdot \epsilon_\tau(e) > \epsilon_{\tau,e}^{max}\right]$ or if $\left[\theta_H \cdot \epsilon_H(e) > \epsilon_{H,e}^{max}\right]$,

    then **refine** $e$.

5. for each group $g$, do:

    if $\left[R_{gl}(g) > \theta_{gl} \cdot R_{gl,g}^{max}\right]$ and if $\left[\theta_\tau \cdot \epsilon_\tau(g) < \epsilon_{\tau,g}^{max}\right]$ and if $\left[\theta_H \cdot \epsilon_H(g) < \epsilon_{H,g}^{max}\right]$,

    then **coarsen** $g$.

---





**Table 1.** Refinement criteria for the adaptive mesh refinement (AMR) simulations.

| Experiment | Label | Criterion |
|---|---|---|
| MISMIP3d | AMR R5 | distance of 5 km to GL |
| MISMIP3d | AMR R10 | distance of 10 km to GL |
| MISMIP3d | AMR R15 | distance of 15 km to GL |
| MISMIP+ | AMR R5 | distance of 5 km to GL |
| MISMIP+ | AMR R15 | distance of 15 km to GL |
| MISMIP+ | AMR R30 | distance of 30 km to GL |
| MISMIP+ | AMR ZZ | ZZ error estimator for $\tau$ |

GL = grounding line. $\tau$ = deviatoric stress tensor. The distance to GL
refers to the region with the highest level of refinement. For example,
AMR R5 means that 5 km on both sides of the GL (upstream and
downstream) are refined with the highest level.





**Table 2.** Levels of refinement tested in the experiments.

| Experiment | Level   | Label | Resolution |
|------------|---------|-------|------------|
| MISMIP3d   | 0 (CM)  | L0    | 5 km       |
| MISMIP3d   | 1       | L1    | 2.5 km     |
| MISMIP3d   | 2       | L2    | 1.25 km    |
| MISMIP3d   | 3       | L3    | 625 m      |
| MISMIP+    | 0 (CM)  | L0    | 4 km       |
| MISMIP+    | 1       | L1    | 2 km       |
| MISMIP+    | 2       | L2    | 1 km       |
| MISMIP+    | 3       | L3    | 500 m      |
| MISMIP+    | 4       | L4    | 250 m      |

CM = coarse mesh, common for Bamg and NeoPZ.





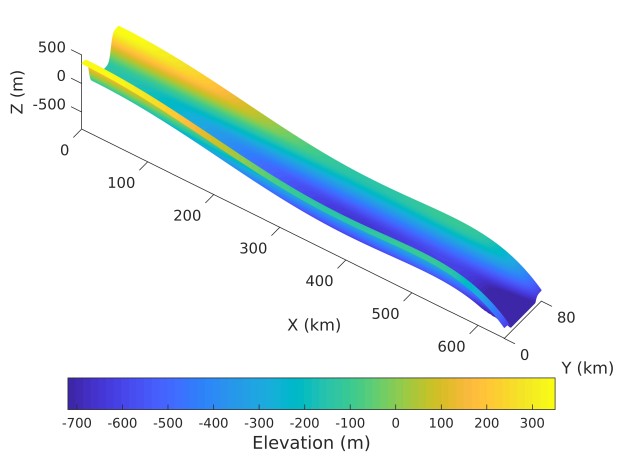

**Figure 3.** The bedrock topography for the MISMIP+ experiment (Asay-Davis et al., 2016).





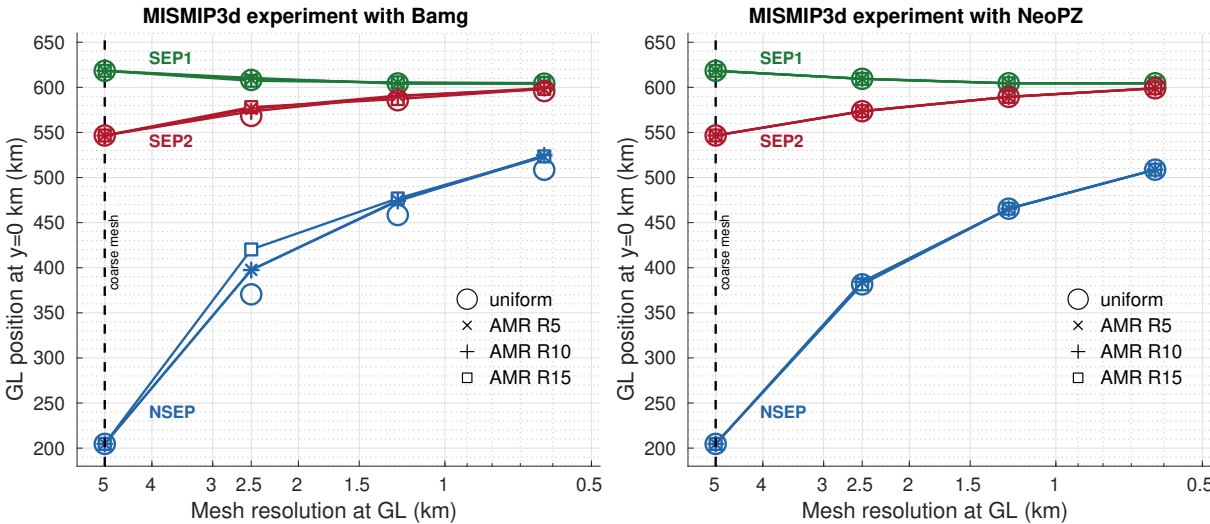

**Figure 4.** Grounding line (GL) positions at steady state obtained from the coarse mesh and from adaptive mesh refinement (AMR) using the refinement criterion based on the element distance to GL, $R_{gl}$. Three element distances are used and compared: $R_{gl} = 5$, $= 10$ and $= 15$ km. The meshes generated with these distances are labeled as AMR R5, AMR R10 and AMR R15, respectively (see Tables 1 and 2). Results from uniformly refined meshes (labeled as uniform) are also shown. The simulations are carried out through the mesh generators Bamg (left) and NeoPZ (right) using 3 sub-element parameterizations: NSEP, SEP1 and SEP2.



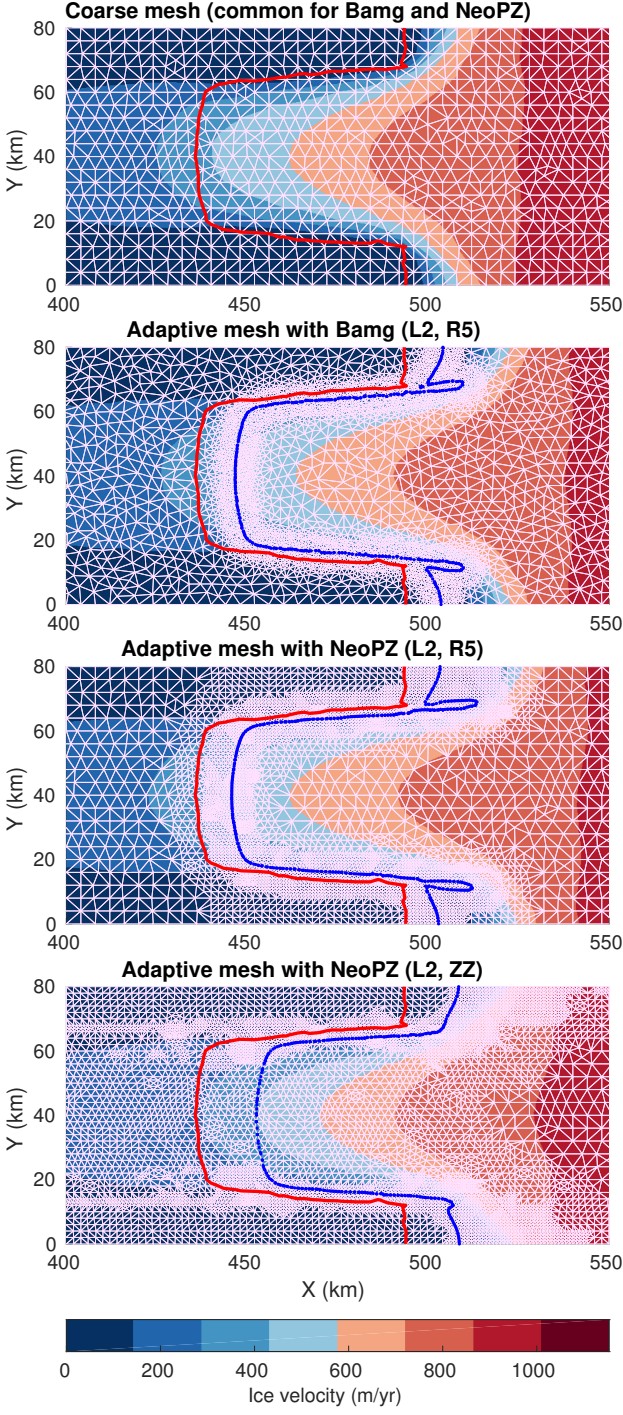

**Figure 5.** Examples of adaptive meshes for MISMIP+ experiment using different refinement criteria and mesh generators (see Tables 1 and 2). **Red line**: grounding line position at steady state obtained with the coarse mesh. **Blue dots**: grounding line position at steady state obtained with each adaptive mesh.





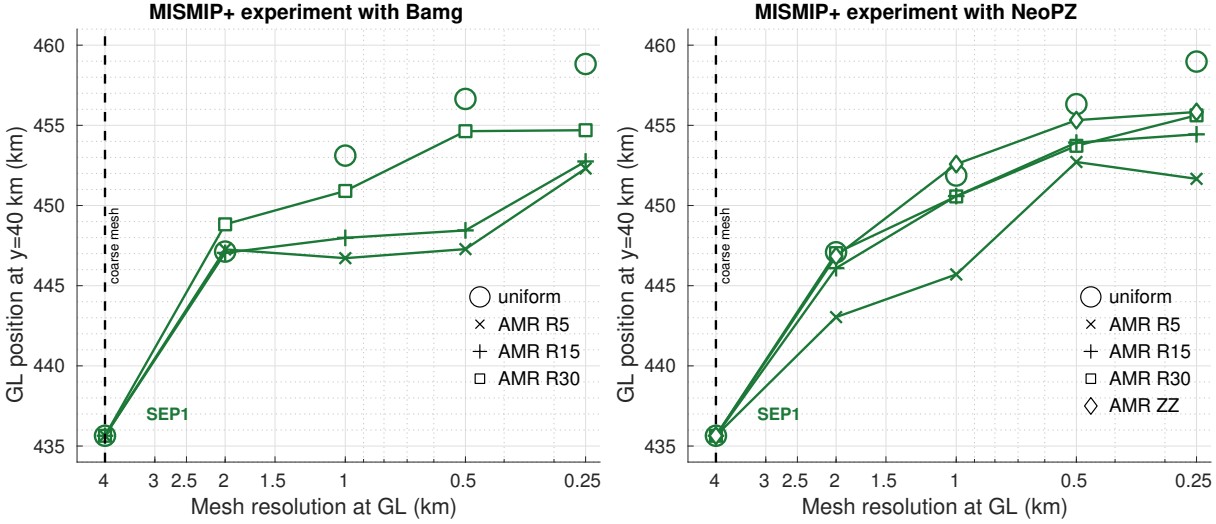

**Figure 6.** Grounding line (GL) positions at steady state obtained from the coarse mesh and from adaptive mesh refinement (AMR) for 4 refinement criteria: R5, R15, R30 and ZZ (see Tables 1 and 2). Results from uniformly refined meshes (uniform) are also shown. The simulations are carried out through the mesh generators Bamg (left) and NeoPZ (right) using sub-element parameterization type 1 (SEP1).





**Figure 7.** Spatial distribution of the ZZ error estimator in the coarse and adaptive meshes (AMR) used in the MISMIP+ experiments. The ZZ error values are normalized between 0 and 1. Black lines are the grounding line positions at steady state obtained with the respective meshes. The adaptive meshes shown here are generated by NeoPZ and the criteria used (R5 and ZZ) are summarized in Table 1.



**Table 3.** AMR time performance comparison for the experiment Ice1r, MISMIP+.

| Level ref. | CPU time (s) | Nb elem. | GL pos. (km) |
|---|---|---|---|
| L0 uniform | 11 | 6,780 | 376.273 |
| L1 uniform | 50 | 27,706 | 384.734 |
| L2 uniform | 216 | 107,722 | 391.513 |
| L3 uniform | 1121 | 473,446 | 394.100 |
| L4 uniform | 2663 | 1,780,012 | 394.667 |
| L3 AMR R30 (Bamg) | 413 | 112,605 | 393.684 |
| L3 AMR R30 (NeoPZ) | 444 | 134,779 | 393.543 |
| L3 AMR ZZ $\tau$ + R5 | 186 | 46,176 | 393.420 |
| L3 AMR ZZ $\tau$, $H$ | 284 | 51,698 | 395.007 |

Level ref. = level of refinement. Nb elem. = number of elements. GL pos. = grounding line position at the end of the experiment. AMR ZZ $\tau$ + R5 = combination of the criteria ZZ error estimator (deviatoric stress tensor, $\tau$) and element distance to GL ($R_{gl}$ = 5 km, R5). AMR ZZ $\tau$, $H$ = ZZ error estimator calculated for both deviatoric stress tensor and ice thickness.