# Peer review of "Implementation and Performance of Adaptive Mesh Refinement in the Ice Sheet System Model (ISSM v4.14)"

_Geoscientific Model Development, 2018_

## Referee Comment (RC1) · Anonymous Referee #1 · 17 Oct 2018

The paper entitled "Implementation and Performance of Adaptive Mesh Refinement in the Ice Sheet System Model (ISSM v4.14)" by Thiago Dias dos Santos et al., presents the implementation of adaptive mesh refinement (AMR) in the finite element ice flow model ISSM.

The performances of the implementation in terms of accuracy and computing time are assessed using the setups of two recent marine ice sheet intercomparison exercises. Accurate modelling of the grounding line dynamics is particularly important to assess the dynamical contribution of the Antarctic Ice-Sheet to sea-level rise. The recent marine ice sheet model intercomparisons (MISMIPs) have confirmed the sensitivity of the

results to the mesh resolution. Saving numerical resources while maintaining accuracy is then essential for simulations at the scale of the whole ice-sheet. Thus this paper address an interesting question for the community.

I find the paper well written and the experiments clearly described and discussed, so I have no general nor specific comments. Previous works on the subjects are clearly detailed in the introduction. Many models use an heuristic criterion based on the distance to the grounding line to prescribe areas of high mesh resolution. Here, this heuristic is compared with the results obtained using an error estimator. It is shown that for complex bed topographies the results based on the distance-to-GL criterion are also sensitive to the distance of refinement, advocating for the use of proper error estimators as refinement criteria. This result is important to motivate further theoretical work on such estimators. The implications for real bed topographies (usually no smooth and not noise free) are discussed in the last section and will motivate further numerical studies.

---

## Referee Comment (RC2) · D. Martin (Referee) · 30 Oct 2018

**Overview**

Adaptive mesh refinement (AMR) is a useful numerical tool in ice sheet modeling due to the wide range of resolution requirements, particularly for accurately modeling marine ice sheet dynamics. A large body of evidence points toward the need for very fine model resolution concentrated near grounding lines (GLs), which can migrate over hundreds of kilometers in long-time simulations. AMR provides the capability to efficiently deploy the required fine resolution adaptively only where needed, resulting in

potentially large savings in computational resource requirements. This paper describes an approach taken to add adaptive mesh refinement to the finite-element-based ISSM ice sheet model. The authors present a description of the algorithm in use, including examples incorporating two different pre-existing remeshing packages. The paper is fairly well-written and quite readable, although could use some editing to fix some English-grammar issues. It does a fairly good job of representing the details of the approach taken and represents a useful addition to the literature. I support publication after some relatively minor changes.

The movies in the supplement are very short and fast, making it hard to see the details of what is happening. I'd suggest slowing them down somewhat (maybe by a factor of 2 or so?). They do look impressive, however.

Another general comment – all of the results are presented in terms of GL position, which can be a bit tricky to work with. I'd suggest supplementing with an integrated quantity like ice volume, grounded area, or volume above flotation if that presents a clearer picture.

**Specific comments**

1. p1, line 9: You say here that you use different combinations of the two error estimators, but I didn't see any combinations in the paper – the examples appeared to use *either* distance from the GL or the ZZ error estimator, but never both. In practice, we've found that the combination of the two works best (although as is pointed out elsewhere in the text, we generally use the undivided Laplacian of the velocity field as a proxy for the truncation error).

2. p2, line 10: You might also want to mention the role of ice-shelf buttressing for completeness.

3. p2, line 16: I'd suggest changing "high grid resolution" to "sufficiently high grid resolution" (or "sufficiently fine") – the whole point here is to (locally) apply sufficient resolution to resolve the dynamics in play. What is "sufficient" depends on where you are.

4. p3, line 5: BISICLES is actually at its heart a 2D model, although "$2\frac{1}{2}D$" might be more descriptive due to the vertical reconstruction entailed by the Schoof-Hindmarsh scheme along with the fully 3D temperature/enthalpy discretization.

5. p4, line 20: It would be helpful if you could add a figure showing examples of how the mesh refinement occurs for the two schemes (start with a coarse base mesh, then show how the mesh is evolved in a picture or two). It's too hard to see that level of detail in your figures and animations. I think I can imagine it, but a picture would be helpful here.

6. p4, line 21: "numerical perturbations" – what you're really talking about is "numerical *errors*" (so it would be good to say that explicitly) introduced due to interpolation. It's important to call this out as a source of error that is introduced by the AMR scheme. The reality is that AMR schemes introduce errors; the goal is to ensure that these errors are outweighed by the improved accuracy and efficiency. The fact that you're able to minimize these by minimizing the need to interpolate from old fine-mesh region to new fine-mesh region as the mesh evolves is an important advantage of your approach.

7. p5, line 21: Are you saying that you are reducing all of the interpolated fields (thickness, bedrock, etc) to processor 0 and then broadcasting them all out? If so, this all-to-one reduction will eventually overwhelm you as you push to larger problems at much higher concurrencies. If that's not what you're doing, please clarify which fields you're talking about here.

8. p7, line 9: I suspect that the ZZ error estimator will be much more useful when applied to either the velocity field or the stress tensor, since this will indicate where there is instantaneous error in the dynamics (which is what will be improved via AMR). Since ice thickness is essentially a time-integral of the divergence of the velocity field, applying an error estimator to the thickness field is likely to indicate where errors have accumulated over time; adding refinement in those regions is likely too late.

9. p10, line 23: Which is correct? You should use a uniform (very-)fine-mesh solution as a comparison.

10. p11, line 14: Stability isn't the only reason to refine your timestep – assuming your time integration scheme is consistent, you should also see an $O(\Delta t)$ or possibly $O(\Delta t^2)$ component of the error, which can become important if you refine only spatially without a corresponding reduction in the timestep.

11. p11, line 19: If I understand the approach suggested here, I think the proposal is to initialize the model using a uniform coarse-resolution solution, and then turn on AMR and add resolution as you begin to evolve the ice sheet (and justify it by stating that's how one begins a realistic simulation based on observations). This is almost certain to produce numerical artifacts due to the sudden change in the mesh setup. We have found that it's important to initialize the model (including data assimilation, inverse-problem solution, etc) with at least the resolution configuration that the model will start with to produce an initial condition with as few numerical artifacts as possible (and you actually essentially make that point in the next sentence about requiring both the steady-state and perturbation experiments be carried out using AMR). I realize that this particular sentence only applies to the performance experiments, but it represents a bad idea which might be misconstrued as a suggestion. Why doesn't it make sense to simply run each mesh configuration forward from its own self-consistent steady state? That would be more representative of how one would initialize the model in a realistic configuration.

12. section 4.2: There is one more figure which would be very useful here to supplement Figure 3 – a plot which shows both element counts and solution time for each case, each normalized against the values for the equivalent uniform fine-mesh solutions, on the same graph (along the lines of Figure 21 in Martin and Colella, JCP, 2000). This has the advantage of showing both the relative savings due to AMR while also illustrating the overhead due to AMR – as represented by the gap between the cell counts and execution times. (If fully computing the uniform fine-mesh solution is too expensive, you can likely compute a few timesteps and extrapolate).

13. p12, line 6: Using estimates of the error is also commonly used in AMR models in the community. For example (as you mention elsewhere), BISICLES often uses the Laplacian of the velocity field as an estimate of the truncation error (in a second-order discretization). Others do as well, but I don't have them at hand. (You seem to imply that using an error-estimator is novel)

14. p13, line 3: I think you can make a stronger statement than "an error estimator may be more appropriate" here – I think you could say "is more appropriate", or, if you'd prefer "is likely more appropriate". This is a nice example of why it is important to understand the error structure of your problem when constructing refinement criteria.

15. p13, line 25: Is there a reason you don't demonstrate an example which uses the combination of refinement criteria that you recommend here?

16. p13, line 35: In general, actual savings due to AMR are fairly problem-dependent.

17. p.14, line 12: "mainly in setups where bedrock induces strong buttressing" – I'm not sure that's really all that relevant here. The additional error (and hence
the additional refinement) occurred due to topographical features which induced complicated stress distributions unrelated to the buttressing.

18. Algorithm 1: It appears that in your algorithm, the initial timestep is taken on the initial coarse mesh before any refinement is implemented. Is that the case? If so, then you have the problem that the initial coarse-resolution timestep can contaminate the solution with initial-time errors. In general, as I mentioned earlier, you're a lot better off if you initialize the problem with the AMR mesh you're going to be running with.

19. Figure 5: This figure suffers from the choice of colormaps. White lines are hard to see, particularly in the more highly-refined regions, and the choice of blue dots to represent the AMR GL position makes that hard to discern from the blue velocity colormap. The small size of the plots only adds to the difficulty. What about potentially showing the mesh lines colored by the velocity field colormap, and then using entirely different colors for the grounding lines?

20. Figure 6. This figure is central to the entire effort, and raises a few questions/issues. I find quite a bit puzzling here:

    (a) Why is the ZZ error estimator only being used for the NeoPZ case? Is it only available for the NeoPZ runs? (I couldn't find any statement to that effect in the paper, although it's possible I overlooked it).

    (b) Would it be possible to use different line types (or possibly different colors) to distinguish the different lines? Seeing how the different cases follow the uniform-mesh case is part of the goal (as distinct from looking at individual data points), and that's hard to separate out when all of the lines are identical, particularly for the finer-resolution parts of the NeoPZ plot.

    (c) Do you have any idea why R30 improves continuously woth increasing resolution for the NeoPZ runs, but stagnates for the Bamg runs? Can you apply the error estimator to see what's going on there?

(d) The drop-off of the AMR-ZZ case at the finest resolution for the NeoPZ case is problematic, because it *could* represent a saturation of the ability for AMR to improve the accuracy of your solutions and provide fine-grid accuracy. There are a number of reasons why that might be the case, some of which would indicate potentially serious limitations on your scheme. One possibility, as mentioned earlier, is that temporal errors are beginning to dominate, due to the constant timestep across all resolutions. I suspect, however, that what's happening here is the saturation of your error indicator. The error threshold for refinement should scale appropriately (proportional to $dx^2$, perhaps?) to match the target resolution. If you're using the same numerical value in your error-tagging criterion for all runs (you don't actually mention in the paper how you're choosing that parameter – you should say that), it will act as a switch and turn off once the solution is accurate enough to match the criterion. Trying to ask for more refinement after that won't actually add much, and so the solution improvement will stall. If that's the case, then there are two potentially useful outcomes:

    i. Tightening the refinement criterion for the 0.25km case will improve the accuracy of that result and reduce or eliminate the stalling apparent in figure 6, and

    ii. The corresponding *loosening* of the refinement criterion for coarser cases will potentially reduce the number of refined cells without a corresponding effect on the solution accuracy (improving the computational efficiency of the AMR scheme)

21. Figure 7: What does the error look like for uniform fine-mesh solutions? Also, the white mesh lines are even harder to see here than they were in Figure 5.

**Technical corrections**

1. p1, line 4: "of grounding line" → "of a grounding line"

2. p1, line 6: "adaptive mesh refinement approach, AMR" → "adaptive mesh refinement (AMR) approach"

3. p1, line 9, elsewhere: "MISMIP3d setup" → "the MISMIP3d setup"

4. p2, line 4: "the collapse of WAIS is based on" suggest replacing with "projections of the collapse of WAIS are based on..."

5. p2, line 20: "flux condition at GL..." → "a flux condition at the GL.."

6. p2, line 24: "allows to apply resources..." → "allows resources to be applied..."

7. p3, line 11: "ice flow Elmer/Ice" → "ice flow model Elmer/Ice"

8. p4, line 25: "high adaptive" → "highly adaptive"?

9. p5, line 4: "transient simulation" → "transient simulations"?

10. p5. line 23: "being $\rho_i$ the ice density" → something like "with $\rho_i$ the ice density..."

11. p5, line 24: "vertical plane view" – I'd suggest "vertical cross-section" (also in the caption for fig. 1)

12. p6, line 1: "of GL" → "of the GL"

13. p12, line 10: "what is useful..." – do you mean "which is useful..." (if not, then I'm not sure what this means)

14. p12, line 15: "scheme" → "schemes"

[Figure]

15. p.14, line 1: "even with a hundred..." → "even with hundreds..."

16. p.14, line 7: "implemented here dynamic..." → "implemented dynamic..."

17. p14, line 23: "compenting" → "competing"

———————————————————

---

## Author Comment (AC1) · 5 Dec 2018

On behalf of my co-authors, I attached three (3) files in the zip file: the author's comments, the manuscript with the differences to original highlighted, and the new manuscript version. Attached are also the updated movies.

Please also note the supplement to this comment: https://www.geosci-model-dev-discuss.net/gmd-2018-194/gmd-2018-194-AC1-supplement.zip

2018.

---

## Author Response (AR2)

**Author's responses for the paper "Implementation and Performance of Adaptive Mesh Refinement in the Ice Sheet System Model (ISSM v4.14)"**

Thiago Dias dos Santos, Mathieu Morlighem, Hélène Seroussi, Philippe Remy Bernard Devloo and Jefferson Cardia Simões

December 11th

**Response notes**

We appreciate the comments and the suggestions given by the reviewers and by the Topical Editor. We reproduced below all their comments.

Attached are:

>**Pages 2 to 12:** author's comment to the reviewers;
>**Pages 13 to 49**: manuscript highlighting the modifications suggested by the reviewers;
>**Pages 50 to 53**: author's comment to the Topical Editor;
>**Pages 54 to 90:** manuscript highlighting the modifications suggested by the Topical Editor.

**Author's responses for the paper "Implementation and Performance of Adaptive Mesh Refinement in the Ice Sheet System Model (ISSM v4.14)"**

Thiago Dias dos Santos, Mathieu Morlighem, Hélène Seroussi, Philippe Remy Bernard Devloo and Jefferson Cardia Simões

December 5th

**Response notes**

We appreciate the comments and the suggestions given by the reviewers. We reproduced below all their comments. **The author's responses are in blue and bold**.

Attached are:
> **(a)** the original manuscript highlighting the modifications, and
> **(b)** the new manuscript version.

**Note 1:** In **(a)**, the deleted words or phrases are in red and in a strikethrough way. **Words or phrases inserted in the manuscript are in blue and bold** (e.g.,  **schemes**).

**Note 2**: all modifications are referred to the file **(a)**. Example: **(AMR) approach** , line 6, page 1 (in file (a)).
* * *
**1) Response to Anonymous Referee #1 (RC1)**

The paper entitled "Implementation and Performance of Adaptive Mesh Refinement in the Ice Sheet System Model (ISSM v4.14)" by Thiago Dias dos Santos et al., presents the implementation of adaptive mesh refinement (AMR) in the finite element ice flow model ISSM.

The performances of the implementation in terms of accuracy and computing time are assessed using the setups of two recent marine ice sheet intercomparison exercises. Accurate modelling of the grounding line dynamics is particularly important to assess the dynamical contribution of the Antarctic Ice-Sheet to sea-level rise. The recent marine ice sheet model intercomparisons (MISMIPs) have confirmed the sensitivity of the results to the mesh resolution. Saving numerical resources while maintaining accuracy is then essential for simulations at the scale of the whole ice-sheet. Thus this paper address an interesting question for the community.

I find the paper well written and the experiments clearly described and discussed, so I have no general nor specific comments. Previous works on the subjects are clearly detailed in the introduction. Many models use an heuristic criterion based on the distance to the grounding line to prescribe areas of high mesh resolution. Here, this heuristic is compared with the results obtained using an error estimator. It is shown that for complex bed topographies the results based on the distance-to-GL criterion are also sensitive to the distance of refinement, advocating for the use of proper error estimators as refinement criteria. This result is important to motivate further theoretical work on such estimators. The implications for real bed topographies (usually no smooth and not noise free) are discussed in the last section and will motivate further numerical studies.

**We would like to thank the Anonymous Referee #1 for his/her positive comments and support for publication.**

**2) Response to D. Martin (RC2)**

**Overview**
Adaptive mesh refinement (AMR) is a useful numerical tool in ice sheet modeling due to the wide range of resolution requirements, particularly for accurately modeling marine ice sheet dynamics. A large body of evidence points toward the need for very fine model resolution concentrated near grounding lines (GLs), which can migrate over hundreds of kilometers in long-time simulations. AMR provides the capability to efficiently deploy the required fine resolution adaptively only where needed, resulting in potentially large savings in computational resource requirements. This paper describes an approach taken to add adaptive mesh refinement to the finite-element-based ISSM ice sheet model. The authors present a description of the algorithm in use, including examples incorporating two different pre-existing remeshing packages. The paper is fairly well-written and quite readable, although could use some editing to fix some English-grammar issues. It does a fairly good job of representing the details of the approach taken and represents a useful addition to the literature. I support publication after some relatively minor changes.
**We thank D. Martin for his positive comments and careful reviews. He pointed out important details, and his comments helped to improve the manuscript.**

The movies in the supplement are very short and fast, making it hard to see the details of what is happening. I'd suggest slowing them down somewhat (maybe by a factor of 2 or so?). They do look impressive, however.
**We apologize for the very short and fast movies. We tried to avoid uploading very large files. We slowed them down and we also improved the resolutions of the files.**

Another general comment – all of the results are presented in terms of GL position, which can be a bit tricky to work with. I'd suggest supplementing with an integrated quantity like ice volume, grounded area, or volume above flotation if that presents a clearer picture.
**We added plots with ice volume above flotation (VAF), since this is an integrated quantity that represents potential sea level. The values of VAF depends on the grounding line positions. Therefore, the same AMR mesh-dependency is observed in the VAF plots. Please, see Figure 5, page 31, and Figure 7, page 33.**
**Also, see**
**- page 13, line 31**
**- page 11, line 13**
**- page 10, line 10**
**- page 10, line 19**

**Specific comments**
1. p1, line 9: You say here that you use different combinations of the two error estimators, but I didn't see any combinations in the paper – the examples appeared to use either distance from the GL or the ZZ error estimator, but never both. In practice, we've found that the combination of the two works best (although as is pointed out elsewhere in the text, we generally use the undivided Laplacian of the velocity field as a proxy for the truncation error).
**This is a good point, we actually did not run the combination of the criteria, since the AMR ZZ produced accurate results in comparison to uniform meshes. We decided to run AMR R5+ZZ and we inserted the results in a table, comparing with AMR ZZ and uniform meshes. Please, see Table 4, page 37.**

2. p2, line 10: You might also want to mention the role of ice-shelf buttressing for completeness.
**Agreed. Please, see page 2, line 10.**

3. p2, line 16: I'd suggest changing "high grid resolution" to "sufficiently high grid resolution" (or "sufficiently fine") – the whole point here is to (locally) apply sufficient resolution to resolve the dynamics in play. What is "sufficient" depends on where you are.

**Done, page 2, line 17.**

4. p3, line 5: BISICLES is actually at its heart a 2D model, although "2 1/2 D" might be more descriptive due to the vertical reconstruction entailed by the Schoof-Hindmarsh scheme along with the fully 3D temperature/enthalpy discretization.

**We thank the reviewer for this information. We changed to "2 1/2 D".**
**Please, see page 3, line 6.**

5. p4, line 20: It would be helpful if you could add a figure showing examples of how the mesh refinement occurs for the two schemes (start with a coarse base mesh, then show how the mesh is evolved in a picture or two). It's too hard to see that level of detail in your figures and animations. I think I can imagine it, but a picture would be helpful here.

**We added a figure showing the coarse mesh and the AMR meshes using Bamg and NeoPZ. We also slowed down the animations, and we also changed the other figures according to reviewer suggestions (please, see text below).**
**Please, see Figure 3, page 25.**
**Also, see the cross-reference: page 4, line 26, and page 5, line 2.**

6. p4, line 21: "numerical perturbations" – what you're really talking about is "numerical errors" (so it would be good to say that explicitly) introduced due to interpolation. It's important to call this out as a source of error that is introduced by the AMR scheme. The reality is that AMR schemes introduce errors; the goal is to ensure that these errors are outweighed by the improved accuracy and efficiency. The fact that you're able to minimize these by minimizing the need to interpolate from old fine-mesh region to new fine-mesh region as the mesh evolves is an important advantage of your approach.

**Agreed. We changed that term on page 4, line 23 and line 32.**

7. p5, line 21: Are you saying that you are reducing all of the interpolated fields (thickness, bedrock, etc) to processor 0 and then broadcasting them all out? If so, this all-to-one reduction will eventually overwhelm you as you push to larger problems at much higher concurrencies. If that's not what you're doing, please clarify which fields you're talking about here.

**Your interpretation is correct, we reduce all fields in CPU #0, and then we broadcast to the other CPUs, such that each CPU takes care of the interpolation process related to their respective nodes/elements. This is done because the mesh partition changes every time AMR is called (the goal is to keep a balanced number of elements among partitions). Then, to interpolate the fields from the old mesh to the new adapted mesh, CPU #0 broadcasts the fields to the others; and each CPU performs the interpolation in its own partition.**

**The reviewer is right, this all-to-one reduction may be a bottleneck for large models. We optimized this by doing only one big MPI call, compacting the data structure and avoiding multiple MPI communications. Another way to overcome this problem is to perform AMR each X time steps, where X is a user-definition parameter ("AMR frequency", in the ISSM interface). Just to have an idea about the size of the MPI data transmission, we show a simple calculation. A mesh with ~100,000 elements has ~50,000 vertices. Then, we have: 50,000 x 8 bytes = 0.4 Mb per interpolated field. This size is relatively small considering the transmission rate of current InfiniBands (>10 Gb/s).**

**It is important to note that we are not advocating the AMR strategy used in ISSM is a procedure that should be applied to any Finite Element code or Ice Sheet model. This is**

**specific for the ISSM architecture. Besides that, our AMR strategy has a minimum impact in the ISSM data structure.**

**We added some explanations about our optimization scheme on page 5, line 24.**

8. p7, line 9: I suspect that the ZZ error estimator will be much more useful when applied to either the velocity field or the stress tensor, since this will indicate where there is instantaneous error in the dynamics (which is what will be improved via AMR). Since ice thickness is essentially a time-integral of the divergence of the velocity field, applying an error estimator to the thickness field is likely to indicate where errors have accumulated over time; adding refinement in those regions is likely too late.

**Yes, we agree with the reviewer that the ZZ error estimator is useful for the elliptic part of the ice sheet problem, the SSA equations in this manuscript. Applying the ZZ estimator to the velocity field or stress tensor is a natural approach. In fact, this estimator was proposed by Zienkiewicz and Zhu (*Int J Numer Meth Eng* 1987) for a linear elasticity problem.**

**We extend the estimator to the ice thickness since this field depends on the bedrock elevation. Basically, the bedrock defines the "vertical geometry" of the problem, which impacts the solutions (ice thickness and velocity field). Then, the ZZ estimator for the ice thickness could be an indicator of areas where the bedrock geometry should be improved in terms of resolution, mainly in transient simulations (where the grounded area changes with time). We suspect that the ZZ for the ice thickness could be also useful if applied together with the other one (ZZ for the deviatoric stress). This should be assessed in the future, for real bedrock elevation, where noise is present.**

**In order to make it clear to the reader, we changed the order on page 7, emphasizing the implementation of the deviatoric stress tensor. We also inserted a note.**
**Please, see page 7, line 12 and line 13.**

9. p10, line 23: Which is correct? You should use a uniform (very-)fine-mesh solution as a comparison.

**The original sentence is correct, because we are referring to Figure 5 (now Figure 6). This figure compares the AMR solutions with the coarse mesh, to show how the GL positions are mesh resolution-dependent and refinement criterion-dependent. But to make it clear to the reader, we inserted the GL positions obtained with the (very-)fine-mesh (L4, uniform).**
**Please, see Figure 6, page 32, and page 11, line 2 to line 7.**

10. p11, line 14: Stability isn't the only reason to refine your timestep – assuming your time integration scheme is consistent, you should also see an $O(\Delta t)$ or possibly $O(\Delta t^2)$ component of the error, which can become important if you refine only spatially without a corresponding reduction in the timestep.

**Yes, we agree. We use an implicit scheme for the ice thickness evolution, then we have an $O(\Delta t)$ in terms of the time-integration error. We use the same time-step in these time performance experiments only for comparison (assuming that all cases should have the same error in the time-integration discretization).**

**We decided to rerun the profiling experiments using the time step required to fulfill the CFL condition (0.2 yr). We apply the same time step to all simulations, such that all cases have the same order in terms of the time-integration error.**
**Please, see page 11, line 31.**

11. p11, line 19: If I understand the approach suggested here, I think the proposal is to initialize the model using a uniform coarse-resolution solution, and then turn on AMR and add resolution as you begin to evolve the ice sheet (and justify it by stating that's how one begins a realistic simulation based on observations). This is almost certain to produce numerical artifacts due to the sudden change in the mesh setup. We have found that it's important to initialize the model (including data assimilation, inverse-problem solution, etc) with at least the resolution configuration that the model will start with to produce an initial condition with as few numerical artifacts as possible (and you actually essentially make that point in the next sentence about requiring both the steady-state and perturbation experiments be carried out using AMR). I realize that this particular sentence only applies to the performance experiments, but it represents a bad idea which might be misconstrued as a suggestion. Why doesn't it make sense to simply run each mesh configuration forward from its own self-consistent steady state? That would be more representative of how one would initialize the model in a realistic configuration.

**Yes, we agree that the initial solution and the initial mesh should be carefully defined using the same AMR mesh applied in the experiments (or at least the same mesh resolution). Our idea was only to compare the processing time with AMR within a tolerance in terms of numerical error.**

**To avoid any misunderstanding, we decided to rerun the time-performance experiments using the same AMR mesh to reach the steady state. We also rephrased the experiment descriptions. Please, see page 12, line 1 to line 11, and Table 3, page 35.**

12. section 4.2: There is one more figure which would be very useful here to supplement Figure 3 – a plot which shows both element counts and solution time for each case, each normalized against the values for the equivalent uniform fine-mesh solutions, on the same graph (along the lines of Figure 21 in Martin and Colella, JCP, 2000). This has the advantage of showing both the relative savings due to AMR while also illustrating the overhead due to AMR – as represented by the gap between the cell counts and execution times. (If fully computing the uniform fine-mesh solution is too expensive, you can likely compute a few timesteps and extrapolate).

**We added a figure along the lines of Figure 21 in Martin and Colella, JCP, 2000, as the reviewer suggested.**
**Please, see Figure 9, page 36, and the note on page 12, line 23.**

13. p12, line 6: Using estimates of the error is also commonly used in AMR models in the community. For example (as you mention elsewhere), BISICLES often uses the Laplacian of the velocity field as an estimate of the truncation error (in a second-order discretization). Others do as well, but I don't have them at hand. (You seem to imply that using an error-estimator is novel)

**We changed that sentence such that it doesn't imply that this is a novel estimator.**
**Please, see page 12, line 30 and line 33.**

14. p13, line 3: I think you can make a stronger statement than "an error estimator may be more appropriate" here – I think you could say "is more appropriate", or, if you'd prefer "is likely more appropriate". This is a nice example of why it is important to understand the error structure of your problem when constructing refinement criteria.

**Good point. Done.**
**Please, see page 14, line 1.**

15. p13, line 25: Is there a reason you don't demonstrate an example which uses the combination of refinement criteria that you recommend here?

**Well, initially we didn't run the combination R5+ZZ because using the results based on ZZ only were virtually similar to uniform meshes. The idea was to show how a simple test with an error estimator produced accurate solutions with fewer elements.**

**Our recommendation is based on the following: we know a priori that applying high resolution around GL would reduce the error caused by the basal friction discretization within the elements. In fact, applying only an error estimator does not guarantee that the elements around the GL are refined. We noted this for the MISMIP+ setup. Interestingly, for the MISMIP+ setup, both AMR R5+ZZ and AMR ZZ produced similar results, which does not mean that it would be the case for real ice sheets. Therefore, for general ice sheets, we suspect that using both criteria (R5+ZZ) should work properly.**

**The results using AMR R5+ZZ are shown in a table. We also added a discussion.**
**Please, see Table 4, page 37.**
**Also, see page 14, line 23 to line 33.**

16. p13, line 35: In general, actual savings due to AMR are fairly problem-dependent.
**Yes, we agree. The original sentence refers to our results. We added a few words to make it clearer.**
**Please, see page 15, line 7.**

17. p.14, line 12: "mainly in setups where bedrock induces strong buttressing" – I'm not sure that's really all that relevant here. The additional error (and hence the additional refinement) occurred due to topographical features which induced complicated stress distributions unrelated to the buttressing.
**Yes, we agree. We added "complex stress distributions", and we also keep strong buttressing.**
**Please, see page 15, line 20.**

18. Algorithm 1: It appears that in your algorithm, the initial timestep is taken on the initial coarse mesh before any refinement is implemented. Is that the case? If so, then you have the problem that the initial coarse-resolution timestep can contaminate the solution with initial-time errors. In general, as I mentioned earlier, you're a lot better off if you initialize the problem with the AMR mesh you're going to be running with.
**Yes, the reviewer is right. The initial mesh is crucial for the first time steps. We also noted this issue. We wrote step 1 in Algorithm 1 thinking only in the initial configuration for the steady state phase of the MISMIP3d and MISMIP+ experiments. But for general and real applications, the initial mesh should be also defined using AMR mesh, as pointed out by the reviewer (or at least at the same resolution used in the AMR meshes).**

**We deleted "coarse" in step 1, Algorithm 1, and we added a footnote with this observation. We also added a new reference as examples of initial setup (Lee et al., Annals of Glaciology, 2015). We also refer to Cornford, et al., JCP, 2013.**
**Please, see Algorithm 1 and the footnote, both on page 26.**

19. Figure 5: This figure suffers from the choice of colormaps. White lines are hard to see, particularly in the more highly-refined regions, and the choice of blue dots to represent the AMR GL position makes that hard to discern from the blue velocity colormap. The small size of the plots only adds to the difficulty. What about potentially showing the mesh lines colored by the velocity field colormap, and then using entirely different colors for the grounding lines?
**The original plots in Figure 5 (now Figure 6) were saved in PDF format, so the thickness of the lines were relative to monitor configurations. We decided to save the figures in PNG format, and we changed them as the reviewer suggested.**
**Please, see Figure 6, page 32.**

20. Figure 6. This figure is central to the entire effort, and raises a few questions/issues. I find quite a bit puzzling here:

(a) Why is the ZZ error estimator only being used for the NeoPZ case? Is it only available for the NeoPZ runs? (I couldn't find any statement to that effect in the paper, although it's possible I overlooked it).

**We have performed these simulations only for NeoPZ, but the ZZ is also available for Bamg. We decided to run with Bamg as well, and we added the results on Figure 6 (now Figure 7). AMR ZZ using Bamg performs similarly to AMR ZZ using NeoPZ.**

**NeoPZ generates nested meshes, which is suitable to perform AMR using error estimators. In nested meshes there is a hierarchy between elements and their refined "children". Then, it is relatively easy to refine (coarsen) elements (group of elements), in the sense of Algorithm 2 in the manuscript.**

**For Bamg, there are different approaches to use estimators, for example: applying the desired mesh resolution in selected nodes (where element errors are high, following Algorithm 2), or applying the error as a 2D metric in the original Bamg algorithm. Here, we applied the desired resolution at the nodes according to the error of the adjacent elements, following Algorithm 2.**
**Please, see Figure 7, page 33.**

(b) Would it be possible to use different line types (or possibly different colors) to distinguish the different lines? Seeing how the different cases follow the uniform-mesh case is part of the goal (as distinct from looking at individual data points), and that's hard to separate out when all of the lines are identical, particularly for the finer-resolution parts of the NeoPZ plot.
**Done. Please, see Figure 7, page 33.**

(c) Do you have any idea why R30 improves continuously with increasing resolution for the NeoPZ runs, but stagnates for the Bamg runs? Can you apply the error estimator to see what's going on there?

**We suspect this is due to our parameter settings in terms of the mesh adaptation. In Bamg, we applied a big ratio between adjacent elements. This creates a "transition zone" between the highest and lowest resolutions. See Figure A below. For NeoPZ, we used a larger "transition zone", see Figure B below. These settings were a simple choice during our simulations (we did not try to find the best parameters for each mesher). Of course, NeoPZ generated more elements in this case compared to Bamg. But, for this case, the ZZ error estimator (deviatoric stress) is higher around the refined zone in Bamg mesh in comparison to the mesh generated by NeoPZ.**

**Also, as we extended the "transition zone" in NeoPZ, the resolution of the bedrock elevation increased more in comparison to the mesh generated by Bamg, since we calculate the MISMIP+ bed elevation in hard code. The error due to the bedrock geometry may be also affecting the results of AMR R30 with Bamg (see further discussion in the next response).**

[Figure]

**Figure A, AMR R30 using Bamg.**

[Figure]

**Figure A, AMR R30 using NeoPZ.**

(d) The drop-off of the AMR-ZZ case at the finest resolution for the NeoPZ case is problematic, because it could represent a saturation of the ability for AMR to improve the accuracy of your solutions and provide fine-grid accuracy. There are a number of reasons why that might be the case, some of which would indicate potentially serious limitations on your scheme. One possibility, as mentioned earlier, is that temporal errors are beginning to dominate, due to the constant timestep across all resolutions. I suspect, however, that what's happening here is the saturation of your error indicator. The error threshold for refinement should scale appropriately (proportional to dx 2 , perhaps?) to match the target resolution. If you're using the same numerical value in your error-tagging criterion for all runs (you don't actually mention in the paper how you're choosing that parameter – you should say that), it will act as a switch and turn off once the solution is accurate enough to match the criterion. Trying to ask for more refinement after that won't actually add much, and so the solution improvement will stall. If that's the case, then there are two potentially useful outcomes:

i. Tightening the refinement criterion for the 0.25km case will improve the accuracy of that result and reduce or eliminate the stalling apparent in figure 6, and

ii. The corresponding loosening of the refinement criterion for coarser cases will potentially reduce the number of refined cells without a corresponding effect on the solution accuracy (improving the computational efficiency of the AMR scheme)

We actually did not mention the values of the error thresholds used in AMR ZZ simulations. We apologize about this.

For simplicity, we use the same error threshold for all levels of refinement. Probably, for the highest level (L4, resolution equal to 250 m), there is some saturation in the ZZ error indication, as the reviewer pointed out. Our goal was not to find the best parameters (error thresholds). Instead, we wanted to show how a simple simulation using an error estimator produced results very close to the uniformly refined meshes, with fewer elements than the ones produced by AMR R30 meshes.

We don't think the drop-off of the AMR-ZZ is caused by temporal errors because we run until a steady state is reached. Although the steady state in the MISMIP+ setup presents some kind of oscillations (see the movies), we don't think the time stepping would impact the "mean" steady state solution. Also, we noted that as the mesh resolution increases, the oscillations tend to vanish.

Then, we think there are other components of the numerical errors in the MISMIP+ setup. There are at least three (3) source of numerical error (which are not necessarily independent of each other):
1) Basal friction discretization within the elements around the GL;
2) High gradient changes in the velocity field near the GL;
3) Bedrock geometry.

(1) is a model-dependent. Using SEP1 and SEP2 the numerical convergence is higher than using NSEP (please, see Figure 5 for MISMIP3d). Despite that, high resolution at the GL is necessary to reduce this error component. The AMR R5 criterion should reduce this error.

(2) is a physical problem-dependent. We know that around the GL, the deviatoric stress tensor dominates the stress balance (e.g., see Schoof, C., JGR 2007). The length of the region where the velocity field changes significantly (where the deviatoric stress tensor dominates) depends on the geometry of the problem (the bedrock geometry). Refining the elements in this region will reduce the error of the deviatoric stress tensor (the derivative of the velocity field).
The AMR ZZ criterion should reduce this component.

(3) is the error due to the geometry discretization. As we mentioned in the original manuscript, the MISMIP+ bed elevation is calculated directly in the code from the input every time AMR is called. It means that the resolution of the MISMIP+ bedrock is increased during the refinement process only in the refined regions. Therefore, not necessarily applying AMR R5 or AMR ZZ (or combination of both) should improve the MISMIP+ bedrock geometry as the uniformly refinement does. This explains why applying AMR ZZ, even tightening the refinement criterion for the 0.25 km(AMR)-mesh (we did the reviewer suggestion, "i"), does not improve the solutions in comparison to the 0.5 km(AMR)-mesh and to the 0.25 km(uniform)-mesh. Probably this error component also affected the result of AMR_R30 using Bamg (0.25 km-resolution, see the last response above).

In fact, increasing the bedrock geometry resolution during the mesh adaptation makes the problem different from a mathematical point of view. But the comparison between AMR and uniform meshes are yet plausible. In this sense, the difference between AMR ZZ and uniform mesh for 0,25 km-resolution is less than 1%.

We stress that more studies should be carried out using real bedrock topographies, to analyze the weight of the component (3) of the numerical errors.

**The thresholds were written in the legends of Figure 7 (page 33), Table 3 (page 35) and Figure 9 (page 36).**

21. Figure 7: What does the error look like for uniform fine-mesh solutions? Also, the white mesh lines are even harder to see here than they were in Figure 5.
**We inserted a figure for the uniform fine-mesh solution (same level of the AMR meshes). We modified the legend accordingly.**

**We decided to remove the white lines (mesh) since the goal of Figure 7 (now Figure 8) is to show the spatial distribution of the ZZ error estimator around the grounding line. Please, see Figure 8, page 34.**

**Technical corrections**

1. p1, line 4: "of grounding line" → "of a grounding line"
**Done. Page 1, line 4.**

2. p1, line 6: "adaptive mesh refinement approach, AMR" → "adaptive mesh refinement (AMR) approach"
**Done. Page 1, line 6.**

3. p1, line 9, elsewhere: "MISMIP3d setup" → "the MISMIP3d setup"
**Done. Page 1, line 10, and elsewhere.**

4. p2, line 4: "the collapse of WAIS is based on" suggest replacing with "projections of the collapse of WAIS are based on..."
**Replaced as suggested. Page 2, line 1.**

5. p2, line 20: "flux condition at GL..." → "a flux condition at the GL.."
**Done. Page 2, line 21.**

6. p2, line 24: "allows to apply resources..." → "allows resources to be applied..."
**Replaced as suggested. Page 2, line 25.**

7. p3, line 11: "ice flow Elmer/Ice" → "ice flow model Elmer/Ice"
**Done. Page 3, line 12.**

8. p4, line 25: "high adaptive" → "highly adaptive"?
**Done. Page 4, line 27.**

9. p5, line 4: "transient simulation" → "transient simulations"?
**Done. Page 5, line 6.**

10. p5. line 23: "being $\rho i$ the ice density" → something like "with $\rho i$ the ice density..."
**Replaced as suggested. Page 6, line 1.**

11. p5, line 24: "vertical plane view" – I'd suggest "vertical cross-section" (also in the caption for fig. 1)
**Replaced as suggested (in both places). Page 6, line 2.**

12. p6, line 1: "of GL" → "of the GL"
**Done. Page 6, line 3.**

13. p12, line 10: "what is useful..." – do you mean "which is useful..." (if not, then I'm not sure what this means)
**Yes, the correct is "which is useful". We corrected that phrase. Page 13, line 4.**

14. p12, line 15: "scheme" → "schemes"
**Done. Page 13, line 9.**

15. p.14, line 1: "even with a hundred..." → "even with hundreds…"
**Done. Page 15, line 8.**

16. p.14, line 7: "implemented here dynamic..." → "implemented dynamic…"
**Done. Page 15, line 15.**

17. p14, line 23: "compenting" → "competing"
**Done. Page 16, line 1.**

[revised manuscript text omitted]

December 10th

**Response notes**

We appreciate the comments and the suggestions given by the Topical Editor, A. Robel. We reproduced below all his comments. **The author's responses are in blue and bold**.

Below is the diff version of the manuscript, highlighting the modification. The figures were removed to avoid large files.
(Page numbers and lines from the diff version)
* * *
**1) Response to A. Robel, Topical Editor**

Comments to the Author:

Thank you for submitting a thorough response to the reviewers and a suitable revised manuscript. As you have adequately responded to all reviewer comments, your manuscript is accepted pending some technical corrections, primarily related to readability. These correction are detailed below.

Best,
Alex Robel

Topical Editor
Geoscientific Model Development
Assistant Professor
Georgia Institute of Technology
* * *
Technical corrections required before final acceptance:
(Page numbers and lines from diff'ed version)

Page 1:
Ln 1: resolution in ice sheet models
**Done**. **Ln 2.**
Ln 9-13: delete sentences starting from "We find…" to "…the complex bedrock topography of MISMIP+" These are quite specific results, and the abstract does not necessarily need them.
**Done. Ln 9-13.**
Ln 15: The ZZ estimator helps
**Done**. **Ln 15.**

Page 2:
Ln 9: Recently, WAIS has
**Done**. **Ln 9.**
Ln 10-11: under ice shelves, reduced the buttressing they provide to inland ice, and trigged the retreat of grounding lines around WAIS observed
**Done. Ln 10-11.**

Ln 14: accuracy of the result is
**Done. Ln 14.**

Ln 18: identify which intercomparison projects
**Done. Ln 18-19.**

Ln 27: as is the case for GL dynamics as defined in Schoof (2007)
(From the editor: To be clear, it is not necessarily the case that the step function in basal friction at the grounding line is necessarily the "right" way to described grounding line dynamics, it just the way that Schoof did it, and so is now commonly done in the field. Do note, however, that when the basal friction transition is smooth, as in Gladstone et al. 2017 in The Cryosphere, the resolution requirements at the grounding line are less stringent. Some have suggested this is a more realistic representation of reality.)
**Done. We aggree with the Topical Editor that the model Schoof (2007) applied (Weertman's model) is not necessarily the "right" way; a Coulomb's model may be a more realistic, for example. But in terms for numerical representation, both models represent a "discontinuity" within the elements (in Coulomb's model it is smoother, wich may reduce the high mesh resolution requirement).**
**Ln 27.**

Page 3:
Ln 17: make it clear here that one or the other mesh generator is used, not both simultaneously
**Done. Ln 18.**

Page 4:
Ln 13: in many cores
**Done. Ln 13.**
Ln 17: of the mesh generators BamG
**Done. Ln 17.**
Ln 25: is the transition zone length specified by the user?
**Actually the "transition zone" here is a "mesh transition zone" (where the elements size changes gradually). This is not the "transition zone" in the context of Pattyn et al. (2006, JGR). Then, to avoid any misunderstanding, we change "transition zone" to "mesh transition zone" everywhere.**
**We also added "by the user". Ln 25.**

Page 5:
Ln 2: of an adapted mesh
**Done. (also changed to "adaptive mesh"). Ln 2.**
Ln 6: transient ISSM simulations
**Done. Ln 6.**
Ln 29: Grounding line dynamics are implemented
**Done. Ln 28.**

Page 6:
Ln 1: flotation height
**Done. Page 5, Ln 31.**

Page 7:
Ln 18: for an explanation of how these
**Done. Ln 14.**

Page 8:
Ln 7: n=3, a commonly used value for the exponent in Glen's flow law
**Done. Ln 4.**
Ln 13: does accumulation rate have to be specified in order to reach a steady state? (I would think yes if there is ice outflow from the domain and it reaches a steady-state)
**Good point. We didn't write the values of the accumulation rate. We apologize for this. Page 8 Ln 10-11, and Page 9, Ln 18-19.**
Ln 19: What about the reviewer comments regarding use of coarse mesh for initialization? Does this present a problem for these simulations?
**Good point. This is not a problem in these experiments because we are running until the steady state. We added a footnote to make it clearer to the reader. Page 8, Ln 18 (and the footnote on Page 8), and Page 9, Ln 23 (and the footnote on Page 9).**
Ln 31: SPE2?
**Corrected. Ln 30.**

Page 10:
Ln 7: the uniformly refined mesh
**Done. Ln 7.**

Page 11:
Ln 3: position differs
**Done. Ln 3.**
Ln 4: position differs
**Done. Ln 4.**
Ln 9: below in what sense?
**Changed to "differ from". Ln 8-9.**
Ln 10: not sure what marginally means here
**Changed to "virtually". Ln 9.**
Ln 25: terms of efficiency
**Done. Ln 24.**

Page 12:
Ln 17-20: you use inconsistent types of numbers to indicate the difference in computational times between these sentences (a % first, then a fraction of, then a factor less than). Perhaps switch to one common way to describe these, like a % of the benchmark non-AMR computational time.
**Done. We switched to %. Ln 9-14.**
Ln 17: L30?
**Corrected (R30). Ln 9.**
Ln 21: is similar, and the ratio
**Done. Ln 15.**
Ln 29: terms of accuracy of the simulated grounding line
**Done. Ln 23.**

Page 13:
Ln 4: step, and is useful
**Done. Page 12, Ln 31.**
Ln 22: ice sheet model
**Done. Ln 16.**

Ln 23: it is the case that these specific results depend on the particular topography and assumptions regarding basal friction across the grounding line.

**Yes, we agree. We changed that sentence specifying the results were valid for MISMIP3d. Ln 16-17.**

Page 14:
Ln 10: implemented in ice sheet models, including those based on
**Done. Ln 4-5.**
Ln 21: there is no small-scale bed topography, the numerical error
**Done. Ln 15.**
Ln 28-29: the extension of the grounding zone (where…
**Done. Ln 22-23.**

Page 15:
Ln 1: The grounding zone is not
**Done. Page 14, Ln 30.**
Ln 18: the grounding zone to reduce
**Actually the "transition zone" here refers to the "mesh transition zone" (zone between the lowest and highest mesh resolution), not to the "grounding zone".**
**To avoid any misunderstanding, we decide to change "transition zone" to "mesh transition zone" along the whole manuscript.**
**Page 15, Ln 13 (and elsewhere).**

[revised manuscript text omitted]

Binary file (standard input) matches